# ES-ENAS: Efficient Evolutionary Optimization for Large-Scale Hybrid Search Spaces

## Abstract

In this paper, we approach the problem of optimizing blackbox functions over large hybrid search spaces consisting of both combinatorial and continuous parameters. We demonstrate that previous evolutionary algorithms which rely on mutation-based approaches, while flexible over combinatorial spaces, suffer from a curse of dimensionality in high dimensional continuous spaces both theoretically and empirically, which thus limits their scope over hybrid search spaces as well. In order to combat this curse, we propose ES-ENAS, a simple and modular joint optimization procedure combining the class of sample-efficient smoothed gradient techniques, commonly known as Evolutionary Strategies (ES), with combinatorial optimizers in a highly scalable and intuitive way, inspired by the *one-shot* or *supernet* paradigm introduced in Efficient Neural Architecture Search (ENAS). By doing so, we achieve significantly more sample efficiency, which we empirically demonstrate over synthetic benchmarks, and are further able to apply ES-ENAS for architecture search over popular RL benchmarks.

## 1 Introduction and Related Work

We consider the problem of optimizing an expensive blackbox function $f : (\mathcal{M}, \mathbb{R}^d) \to \mathbb{R}$, where $\mathcal{M}$ is a combinatorial search space consisting of potentially multiple layers of categorical and discrete variables, and $\mathbb{R}^d$ is a high dimensional continuous search space, consisting of potentially hundreds to thousands of parameters. Such scenarios broadly encompass the space of large *non-differentiable* networks, particularly useful in the thriving field of Automated Reinforcement Learning (AutoRL) (Parker-Holder et al., 2022), where $m \in \mathcal{M}$ represents an architecture specification and $\theta \in \mathbb{R}^d$ represents a collection of possible neural network weights, together to form a policy $\pi_{m,\theta} : \mathcal{S} \to \mathcal{A}$ mapping from search space $\mathcal{S}$ to action space $\mathcal{A}$ in which the goal is to maximize total reward in a given environment.

There have been a flurry of previous methods for approaching complex, combinatorial search spaces, especially in the evolutionary algorithm domain, including the well-known NEAT (Stanley and Miikkulainen, 2002). More recently, the neural architecture search (NAS) community has also adopted a multitude of blackbox optimization methods for dealing with NAS search spaces, including policy gradients via Pointer Networks (Vinyals et al., 2015) and more recently Regularized Evolution (Real et al., 2018). Such methods have been successfully applied to applications ranging from image classification (Zoph and Le, 2017) to language modeling (So et al., 2019), and even algorithm search/genetic programming (Real et al., 2020; Co-Reyes et al., 2021). Combinatorial algorithms allow huge flexibility in the search space definition, which allows optimization over generic spaces such as graphs, but many techniques rely on the notion of zeroth-order *mutation*, which can be

inappropriate in high dimensional continuous space due to large sample complexity (Nesterov and Spokoiny, 2017).

On the other hand, there are also a completely separate set of algorithms for attacking high dimensional continuous spaces $\mathbb{R}^d$. These include global optimization techniques including the Cross-Entropy method (de Boer et al., 2005) and metaheuristic methods such as swarm algorithms (Mavrovouniotis et al., 2017). More local-search based techniques include the class of methods based on Evolution Strategies (ES) (Salimans et al., 2017), such as CMA-ES (Hansen et al., 2003; Krause et al., 2016; Varelas et al., 2018) and Augmented Random Search (ARS) (Mania et al., 2018a). ES has been shown to perform well for reinforcement learning policy optimization, especially in continuous control (Salimans et al., 2017) and robotics (Gao et al., 2020; Song et al., 2020a). Even though such methods are also zeroth-order, they have been shown to scale better than previously believed (Conti et al., 2018; Liu et al., 2019a; Rowland et al., 2018) on even millions of parameters (Such et al., 2017) due to advancements in heuristics (Choromanski et al.,

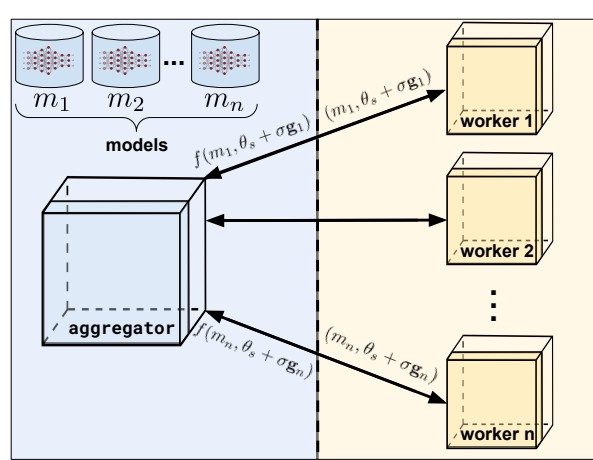

Figure 1: Representation of the ES-ENAS aggregator-worker pipeline, where the aggregator proposes models $m_i$ in addition to a perturbed input $\theta + \sigma \mathbf{g_i}$, and the worker the computes the objective $f(m_i, \theta + \sigma \mathbf{g_i})$, which is sent back to the aggregator. Both the training of the weights $\theta$ and of the model-proposing controller $p_\phi$ rely on the number of worker samples to improve performance.

2019a) and Monte Carlo gradient estimation techniques (Choromanski et al., 2019b; Yu et al., 2016). Unfortunately, these analytical techniques are limited only to continuous spaces and at best, basic categorical spaces via softmax reparameterization.

One may thus wonder whether it is possible to combine the two paradigms in an **efficient** manner. For example, in AutoRL and NAS applications, it would be extremely wasteful to run an end-to-end ES-based training loop for every architecture proposed by the combinatorial algorithm. At the same time, two practical design choices we must strive towards are also **simplicity** and **modularity**, in which a user may easily setup our method and arbitrarily swap in continuous algorithms like CMA-ES (Hansen et al., 2003) or combinatorial algorithms like Policy Gradients (Vinyals et al., 2015) and Regularized Evolution (Real et al., 2018), for specific scenarios. **Generality** is also an important aspect as well, in which our method should be applicable to generic hybrid spaces. For instance, HyperNEAT (Stanley et al., 2009) addresses the issue of high dimensional neural network weights by applying NEAT to evolve a smaller hypernetwork (Ha et al., 2017) for weight generation, but such a solution is domain specific and is not applicable to broader blackbox optimization problems. Similarly restrictive, Weight Agnostic Neural Networks (Gaier and Ha, 2019) do not train any continuous parameters and apply NEAT to only the combinatorial spaces of network structures, and other works (Moriguchi and Honiden, 2012; Miikkulainen et al., 2017) similarly mainly target neural networks specifically. Works that do address blackbox hybrid spaces include Bayesian Optimization (Deshwal et al., 2021) or Population Based Training (Parker-Holder et al., 2021), but only in hyperparameter tuning settings whose search spaces are significantly smaller.

One of the first cases of combining differentiable continuous optimization with combinatorial optimization was from Efficient NAS (ENAS) (Pham et al., 2018), which introduces the notion of *weight sharing* to build a maximal *supernet* containing all possible weights $\theta_s$ where each *child model m* only utilizes certain subcomponents and their corresponding weights from this supernet. Child models $m$ are sampled from a *controller $p_\phi$*, parameterized by some state $\phi$. **The core idea is to perform separate updates to $\theta_s$ and $\phi$ in order to respectively, improve both neural network weights and architecture selection at the same time.** However, ENAS and followup variants (Akimoto et al., 2019) were originally proposed in the setting of using a GPU worker with autodifferentiation over $\theta_s$ in mind for efficient NAS training.

In order to adopt ENAS's joint optimization into the fully blackbox (and potentially non-differentiable) scenario involving hundreds/thousands of CPU-only workers, we introduce the ES-ENAS algorithm, which is practically implemented as a simple add-on to a standard synchronous optimization scheme commonly found in ES, shown in Fig. 1. We explain the approach formally below.

## 2 ES-ENAS Method

**Preliminaries**  In defining notation, let $\mathcal{M}$ be a combinatorial search space in which $m$ are drawn from, and $\theta \in \mathbb{R}^d$ be the continuous parameter or "weights". For scenarios such as NAS, one may define $\mathcal{M}$'s representation to be the superset of all possible child models $m$. Let $\phi$ represent the state of our combinatorial algorithm or "controller", and let $p_\phi$ its current output distribution over $\mathcal{M}$.

### 2.1 Algorithm

We concisely summarize our ES-ENAS method in Algorithm 1. Below, we provide ES-ENAS's derivation and conceptual simplicity of combining the updates for $\phi$ and $\theta$ into a joint optimization procedure.

The optimization problem we are interested in is $\max_{m \in \mathcal{M}, \theta \in \mathbb{R}^d} f(m, \theta)$. In order to make this problem tractable, consider instead, optimization on the smoothed objective:

$$\widetilde{f}_\sigma(\phi, \theta) = \mathbb{E}_{m \sim p_\phi, \mathbf{g} \sim \mathcal{N}(0, I)} \left[ f(m, \theta + \sigma \mathbf{g}) \right] \quad (1)$$

Note that this smoothing defines a particular distribution $P_{m, \theta}$ across $(\mathcal{M}, \mathbb{R}^d)$, and can be more generalized to the rich literature on *Information-Geometric Optimization* (Ollivier et al., 2017), which can be used to derive different variants and update rules of our approach, such as using CMA-ES or other ES variants

---

**Algorithm 1:** Default ES-ENAS Algorithm, with the few additional modifications to allow ENAS from ES shown in blue.

**Data:** Initial weights $\theta$, weight step size $\eta_w$, precision parameter $\sigma$, number of perturbations $n$, controller $p_\phi$.

**while** *not done* **do**

    Sample i.i.d. vectors $\mathbf{g}_1, \ldots, \mathbf{g}_n \sim \mathcal{N}(0, \mathbf{I})$;

    **foreach** $\mathbf{g}_i$ **do**

        Sample $m_i^+, m_i^- \sim p_\phi$

        $v_i^+ \leftarrow f(m_i^+, \theta + \sigma \mathbf{g}_i)$

        $v_i^- \leftarrow f(m_i^-, \theta - \sigma \mathbf{g}_i)$

        $v_i \leftarrow \frac{1}{2}(v_i^+ - v_i^-)$

    **end**

    Update weights $\theta \leftarrow \theta + \eta_w \frac{1}{\sigma n} \sum_{i=1}^{n} v_i \mathbf{g}_i$

    Update controller $p_\phi$ using all $\{(m, v)\}$

**end**

---

(Wierstra et al., 2014; Heidrich-Meisner and Igel, 2009; Krause, 2019) to optimize $\theta$. For simplicity, we use vanilla ES as it suffices for common problems such as continuous control. Our particular update rule is to use samples from $m \sim p_\phi, \mathbf{g} \sim \mathcal{N}(0, I)$ for updating both algorithm components

in an unbiased manner, as it efficiently reuses evaluations to reduce the sample complexity of both the controller $p_\phi$ and the variance of the estimated gradient $\nabla_\theta \widetilde{f}_\sigma$.

### 2.1.1 Updating the Weights

The goal is to improve $\widetilde{f}_\sigma(\phi, \theta)$ with respect to $\theta$ via one step of the gradient:

$$\nabla_\theta \widetilde{f}_\sigma(\phi, \theta) = \frac{1}{2\sigma} \mathbb{E}_{m \sim p_\phi, \mathbf{g} \sim \mathcal{N}(0,I)} \left[ (f(m, \theta + \sigma \mathbf{g}) - f(m, \theta - \sigma \mathbf{g})) \mathbf{g} \right] \tag{2}$$

Note that by linearity, we may move the expectation $\mathbb{E}_{m \sim p_\phi}$ inside into the two terms $f(m, \theta + \sigma \mathbf{g})$ and $f(m, \theta - \sigma \mathbf{g})$, which implies that the gradient expression can be estimated with averaging singleton samples of the form:

$$\frac{1}{2\sigma}(f(m^+, \theta + \sigma \mathbf{g}) - f(m^-, \theta - \sigma \mathbf{g}))\mathbf{g} \tag{3}$$

where $m^+, m^-$ are i.i.d. samples from $p_\phi$, and $\mathbf{g}$ from $\mathcal{N}(0, I)$.

Thus we may sample multiple i.i.d. child models $m_1^+, m_1^- ..., m_n^+, m_n^- \sim p_\phi$ and also multiple perturbations $\mathbf{g_1}, ..., \mathbf{g_n} \sim \mathcal{N}(0, I)$ and update weights $\theta$ with an approximate gradient update:

$$\theta \leftarrow \theta + \eta_w \left( \frac{1}{n} \sum_{i=1}^{n} \frac{f(m_i^+, \theta + \sigma \mathbf{g_i}) - f(m_i^-, \theta - \sigma \mathbf{g_i})}{2\sigma} \mathbf{g}_i \right) \tag{4}$$

This update forms the "ES" portion of ES-ENAS. As a sanity check, we can see that using a constant fixed $m = m_1^+ = m_1^- = ... = m_n^+ = m_n^-$ reduces Eq. 4 to standard ES/ARS optimization.

### 2.1.2 Updating the Controller

For optimizing over $\mathcal{M}$, we update $p_\phi$ by simply reusing the objectives $f(m, \theta + \sigma \mathbf{g})$ already computed for the weight updates, as they can be viewed as unbiased estimations of $\mathbb{E}_{\mathbf{g} \sim \mathcal{N}(0,I)}[f(m, \theta + \sigma \mathbf{g})]$ for a given $m$. Conveniently, we can use common approaches such as

**Policy Gradient Methods:** $\phi$ are differentiable parameters of a distribution $p_\phi$ (usually a RNN-based controller), with the goal of optimizing the smoothed objective $J(\phi) = \mathbb{E}_{m \sim p_\phi, \mathbf{g} \sim \mathcal{N}(0,I)}[f(m; \theta + \sigma \mathbf{g})]$, whose *policy gradient* $\nabla_\phi J(\phi)$ can be estimated by $\widehat{\nabla}_\phi J(\phi) = \frac{1}{n} \sum_{i=1}^{n} f(m_i, \theta + \sigma \mathbf{g_i}) \nabla_\phi \log p_\phi(m_i)$. The ES-ENAS variant can be seen as estimating a "simultaneous gradient" consisting of the two updates over $\theta$ and $\phi$.

**Evolutionary Algorithms:** In this setting, $\phi$ represents the algorithm state, which usually consists of a *population* of inputs $Q = \{(m_1, \theta_1), ..., (m_n, \theta_n)\}$ with corresponding evaluations (slightly abusing notation) $f(Q) = \{f(m_1, \theta_1), ..., f(m_n, \theta_n)\}$. The algorithm performs a *selection procedure* (usually argmax) which selects an individual $(m_i, \theta_i)$ or potentially multiple individuals $T \subseteq Q$, in order to perform respectively, mutation or crossover to "reproduce" and form a new child instance $(m_{new}, \theta_{new})$. Some prominent examples include Regularized Evolution (Real et al., 2018), NEAT (Stanley and Miikkulainen, 2002), and Hill-Climbing (Golovin et al., 2020; Song et al., 2020b).

## 3 Curse of Continuous Dimensionality

One may wonder why simply using original gradientless evolutionary algorithms such as Regularized Evolution or Hill-Climbing over the entire space $(\mathcal{M}, \mathbb{R}^d)$ is not sufficient. Many algorithms such

as the two mentioned use a variant of the $\arg\max$ operation for deciding ascent direction, and only require a mutation operator $(m, \theta) \to (m', \theta')$, where the most common and natural way of continuous mutation is simple additive mutation: $\theta' = \theta + \sigma_{mut}\mathbf{g}$ for some random Gaussian vector $\mathbf{g}$.

The answer lies in efficiency: for e.g. convex objectives, in terms of convergence rate, ES can be $\widetilde{O}(d)$ times more sample efficient than a mutation-based $\arg\max$ procedure such as Hill-Climbing. More formally, we prove the following instructive theorem over continuous spaces (full proof in Appendix E) assuming standard concave/convex optimization settings (Boyd and Vandenberghe, 2004):

**Theorem 1.** *Let $f(\theta)$ be a $\alpha$-strongly concave, $\beta$-smooth function over $\mathbb{R}^d$, and let $\Delta_{ES}(\theta)$ be the expected improvement of an ES update, while $\Delta_{MUT}(\theta)$ be the expected improvement of a batched hill-climbing update, with both starting at $\theta$ and using $B = o(\exp(d))$ parallel evaluations / workers for fairness. Then assuming optimal hyperparameter tuning, $\Delta_{ES}(\theta) = \Omega\left(\frac{\|\nabla f(\theta)\|_2^2}{\beta}\right)$ while $\Delta_{MUT}(\theta) = O\left(\frac{\|\nabla f(\theta)\|_2^2 \log(B)}{\alpha \cdot \left(\sqrt{d} - \sqrt{\log(B)}\right)^2}\right)$, which leads to an improvement ratio of $\frac{\Delta_{ES}(\theta)}{\Delta_{MUT}(\theta)} = \Omega\left(\frac{1}{\kappa}\frac{(\sqrt{d} - \sqrt{\log(B)})^2}{\log(B)}\right)$ where $\kappa = \beta/\alpha$ is the condition number.*

From the above, to achieve the same level of 1-step improvement as ES, a mutation-based approach must use $\Omega(\exp(d))$ evaluations, effectively brute forcing the entire $\mathbb{R}^d$ search space! Since the number of iterations required for convergence is inversely proportional to the improvement ratio (Boyd and Vandenberghe, 2004), this also implies $\frac{\Delta_{ES}(\theta)}{\Delta_{MUT}(\theta)}$ more samples overall are required, which can be a factor of $\widetilde{O}(d)$ when $B$ is subexponential. The above establishes the theoretical explanation over the effect of large $d$. However, this does not cover the case for non-convex objectives, hybrid spaces, or other types of update schemes, all of which may lack possible theoretical analysis, and thus we also experimentally verify this issue below.

### 3.1 BBOB Experiments

We begin by benchmarking over a simple hybridized variant of the common Black-Box Optimization Benchmark (BBOB) (Hansen et al., 2009). We define our hybrid search space as $(\mathcal{M}, \mathbb{R}^{d_{con}})$, where $\mathcal{M}$ consists of $d_{cat}$ categorical parameters, each of which may take feasible values from an unordered set of equally spaced grid points. An input $(m, \theta)$ is then evaluated using the native BBOB function $f$ originally operating on the input space $\mathbb{R}^{d_{cat}+d_{con}}$. We report the average normalized optimality gap $\frac{f^* - \widehat{f^*}}{f^*}$ as common in e.g. (Müller et al., 2021), where $f^*$ and $\widehat{f^*}$ are the true and algorithm's estimated optimums respectively.

The set of original algorithms we use are: Regularized Evolution (Real et al., 2018), NEAT (Stanley and Miikkulainen, 2002), Random Search, Gradientless Descent/Batch Hill-Climbing (Golovin et al., 2020; Song et al., 2020b) and PPO (Schulman et al., 2017) as a policy gradient baseline[1]. To remain fair and consistent, we use the same mutation $(m, \theta) \to (m', \theta')$ across all mutation-based algorithms, which consists of $\theta' = \theta + \sigma_{mut}\mathbf{g}$ for a tuned $\sigma_{mut}$, and uniformly randomly mutating a single categorical parameter from $m$. All algorithms start at the same randomly sampled initial point. More hyperparameters can be found in Appendix A.3 along with continuous optimizer comparisons (e.g. CMA-ES) in Appendix B.

---

[1]Only for categorical parameters as the default implementation for Pointer Networks (Vinyals et al., 2015; Bello et al., 2016) does not include continuous parameters.

In Figure 2, we experimentally demonstrate the severe degradation of vanilla combinatorial evolutionary algorithms compared to their ES-ENAS-modified counterparts. In the first row, when we only evaluate on the continuous space, we verify that the original ES algorithm significantly outperforms the other vanilla algorithms, as $d_{con}$ increases. Similarly, when the space becomes hybridized in the following rows, each ES-ENAS variant will also outperform against its corresponding original algorithm.

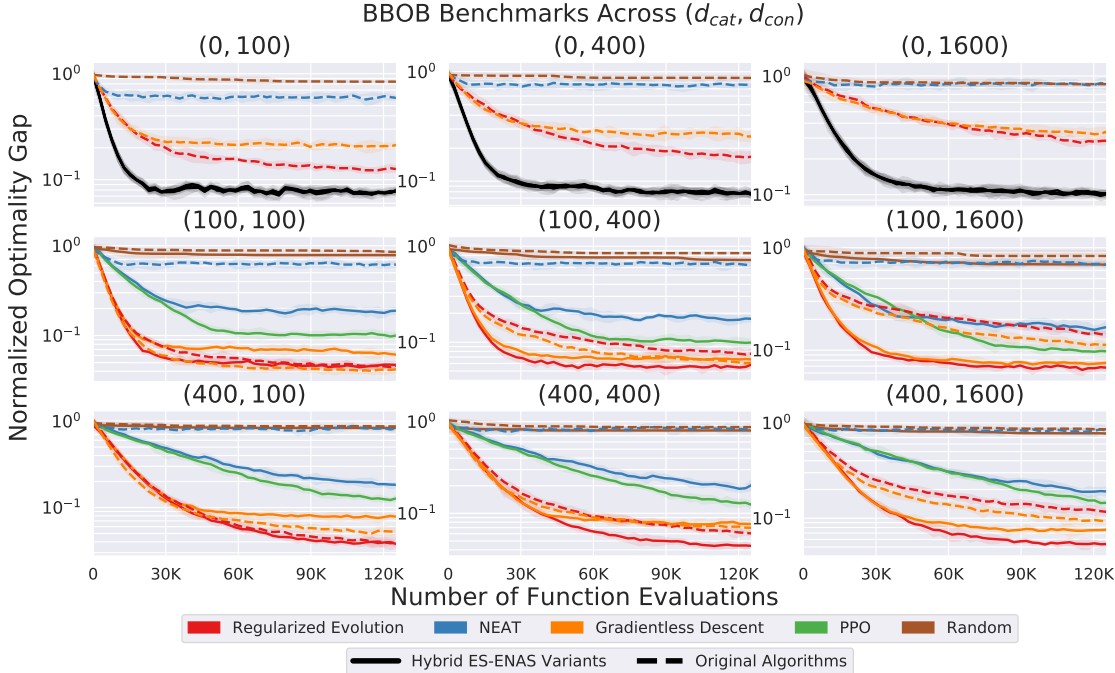

Figure 2: Lower is better. Aggregate performance of every algorithm when ranging $d_{cat}$ and $d_{con}$. **Nearly every original algorithm (dashed line) is used to also create a corresponding ES-ENAS variant (solid line).**

## 4 Neural Network Policy Experiments

In order to benchmark our method over more nested combinatorial structures, we apply our method to two combinatorial problems, **Sparsification** and **Quantization**, on standard Mujoco (Todorov et al., 2012) environments from OpenAI Gym, which are well aligned with the use of ES and also have hundreds to thousands of continuous neural network parameters. Furthermore, such problems are also *reducing parameter count*, which can also greatly improve performance and sample complexity.

Such problems also have a long history, with sparisification methods such as (Rumelhart, 1987; Chauvin, 1989; Mozer and Smolensky, 1989) from the 1980's, Optimal Brain Damage (Cun et al., 1990), regularization (Louizos et al., 2018), magnitude-based weight pruning methods (Han et al., 2015; See et al., 2016; Narang et al., 2017), sparse network learning (Gomez et al., 2019; Lenc et al., 2019), and the recent Lottery Ticket Hypothesis (Frankle and Carbin, 2019). Meanwhile, quantization has been explored with Huffman coding (Han et al., 2016), randomized quantization (Chen et al., 2015), and hashing mechanisms (Eban et al., 2020).

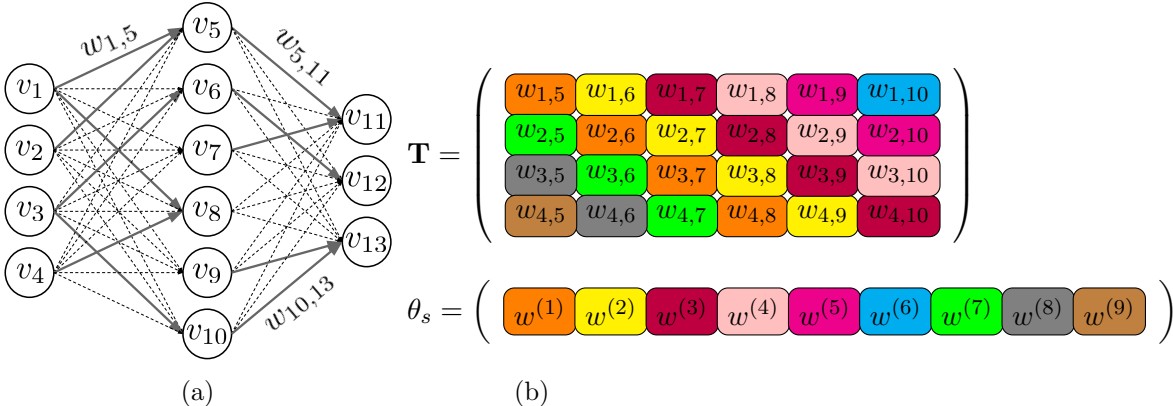

(a)                                              (b)

Figure 3: (a) Example of sparsifying a neural network setup, where solid edges are those learned by the algorithm. (b) Example of quantization using a Toeplitz pattern (Choromanski et al., 2018), for the first layer in Fig. 3a. Entries in each of the diagonals are colored the same, thus sharing the same weight value. The trainable weights $\theta_s = \left(w^{(1)}, ..., w^{(9)}\right)$ are denoted at the very bottom in the vectorized form with 9 entries, which effectively encodes the larger $\mathbf{T}$ with 24 entries.

## 4.1   Setup

We can view a feedforward neural network as a standard directed acyclic graph (DAG), with a set of *vertices* containing values $\{v_1, ..., v_k\}$, and a set of *edges* $\{(i,j)|1 \leq i \leq j \leq k\}$ where each edge $(i,j)$ contains a weight $w_{i,j}$, as shown in Figures 3a and 3b. The goals of sparsification and quantization are to maintain high environment reward while maintaining respectively, a low target number of edges or partitions (for weight sharing). These scenarios possess very large combinatorial policy search spaces (calculated as $|\mathcal{M}| > 10^{68}$, comparable to $10^{49}$ from NASBench-101 (Ying et al., 2019)) that will stress test our ES-ENAS algorithm and are also relevant to mobile robotics (Gage, 2002). Given the results in Subsection 3.1 and since this is a NAS-based problem, for ES-ENAS we use the two most domain-specific controllers, Regularized Evolution and PPO (Policy Gradient) and take the best result in each scenario. Specific details and search space size calculations can be found in Appendix A.4.

## 4.2   Results

As we have already demonstrated comparisons to blackbox optimization baselines in Subsection 3.1, we now focus our comparison to domain-specific baselines for the neural network. These include a DARTS-like (Liu et al., 2019b) softmax *masking* method (Lenc et al., 2019), which applies a trainable boolean matrix mask over weights for edge pruning. We also include strong mathematically grounded baselines for fixed quantization patterns such as Toeplitz and Circulant matrices (Choromanski et al., 2018). In all cases we use the same hyper-parameters, and train until convergence for three random seeds. For masking, we report the best achieved reward with $> 90\%$ of the network pruned, making the final policy comparable in size to the quantization and edge-pruning networks. All results are for feedforward nets with one hidden layer. More details can be found in Appendices C.1 and A.4.

For each class of policies, we compare various metrics, such as the number of weight parameters used, total parameter count compression with respect to unstructured networks, and total number of bits for encoding float values (since quantization and masking methods require extra bits to encode the

partitioning via dictionaries). In Table 1, we see that both sparsification and quantization can be **learned from scratch via optimization using ES-ENAS**, which achieves competitive or better rewards against other baselines. This is especially true against hand-designed (Toeplitz/Circulant) patterns which significantly fail at Walker2d, as well as other optimization-based reparameterizations, such as softmax masking, which underperforms on the majority of environments. The full set of numerical results over all of the mentioned methods can be found in Appendix C.

### 4.3 Neural Network Policy Ablations

In the rest of the experimental section, we provide ablations studies on the properties and extensions of our ES-ENAS method. Because of the nested combinatorial structure of the neural network space (rather than the flat space of BBOB functions), certain behaviors for the algorithm may differ. Furthermore, we also wish to highlight the similarities and differences from regular NAS in supervised learning, and thus raise the following questions:

1. How do controllers compare in performance?
2. How does the number of workers affect the quality of optimization?
3. Can other extensions such as constrained optimization also work in ES-ENAS?

#### 4.3.1 Controller Comparisons

As shown in Subsection 3.1, Regularized Evolution (Reg-Evo) was the highest performing controller when used in ES-ENAS. However, this is not always the case, as mutation-based optimization may be prone to being stuck in local optima whereas policy gradient methods (PG) such as PPO can allow better exploration.

| Env. | Arch. | Reward | # weights | compression | # bits |
|------|-------|--------|-----------|-------------|--------|
| Striker | Quantization | -247 ± 11 | **23** | 95% | 8198 |
| | Edge Pruning | -130 ± 16 | 64 | 93% | **3072** |
| | Masked | -967 ± 200 | 25 | 95% | 8262 |
| | Toeplitz | -129 | 110 | 88% | 4832 |
| | Circulant | **-120** | 82 | 90% | **3936** |
| | Unstructured | **-117 ± 30** | 1230 | 0% | 40672 |
| HalfCheetah | Quantization | **4894 ± 110** | **17** | 94% | 6571 |
| | Edge Pruning | 4016 ± 726 | 64 | 98% | **3072** |
| | Masked | **4806 ± 200** | **40** | 92% | 8250 |
| | Toeplitz | 2525 | 103 | 85% | 4608 |
| | Circulant | 1728 | 82 | 88% | **3936** |
| | Unstructured | 3614 ± 180 | 943 | 0% | 31488 |
| Hopper | Quantization | **3220 ± 119** | **11** | 92% | **3960** |
| | Edge Pruning | **3349 ± 206** | 64 | 84% | **3072** |
| | Masked | 2196 ± 150 | **17** | 91% | 4726 |
| | Toeplitz | 2749 | 94 | 78% | 4320 |
| | Circulant | 2680 | 82 | 80% | 3936 |
| | Unstructured | 2691 ± 201 | 574 | 0% | 19680 |
| Walker2d | Quantization | 2026 ± 46 | **17** | 94% | 6571 |
| | Edge Pruning | **3813 ± 128** | 64 | 90% | **3072** |
| | Masked | 1781 ± 180 | **19** | 94% | 6635 |
| | Toeplitz | 1 | 103 | 85% | 4608 |
| | Circulant | 3 | 82 | 88% | **3936** |
| | Unstructured | **2230 ± 150** | 943 | 0% | 31488 |

Table 1: Comparison of the best policies from six distinct classes of RL networks: Quantization (ours), Edge Pruning (ours), Masked, Toeplitz, Circulant, and Unstructured networks trained with standard ES algorithm (Salimans et al., 2017). Best two metrics for each environment are in **bold**, while significantly low rewards are in red.

We thus compare different ES-ENAS variants, when using Reg-Evo, PG (PPO), and random search (for sanity checking), on the edge pruning task in Fig. 4. As shown, while Reg-Evo consistently converges faster than PG at first, PG eventually may outperform Reg-Evo in asymptotic performance. Previously on NASBENCH-like benchmarks, Reg-Evo consistently outperforms PG in both sample complexity and asymptotic performance (Real et al., 2018), and thus our results on ES-ENAS are surprising, potentially due to the hybrid optimization of ES-ENAS.

Random search has been shown in supervised learning to be a surprisingly strong baseline (Li and Talwalkar, 2019), with the ability to produce even ≥ 80-90 % accuracy (Pham et al., 2018; Real et al., 2018), showing that NAS-based optimization produces most gains ultimately be at the tail end; e.g. at the 95% accuracies. In the ES-ENAS setting, this is shown to occur for easier RL environments such as Striker (Fig. 4) and Reacher (shown in Appendices C.2, C.3). However, for the majority of RL environments, a random search controller is unable to train at all, which also makes this regime different from supervised learning.

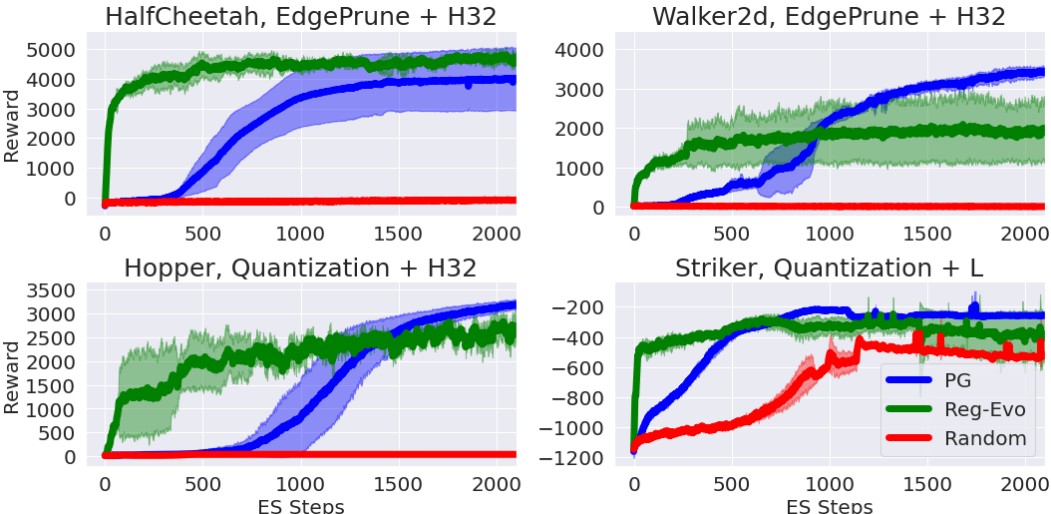

Figure 4: Comparisons across different environments when using different controllers, on the edge pruning and quantization tasks, when using a linear layer (L) or hidden layer of size 32 (H32).

### 4.3.2 Controller Sample Complexity

We further investigate the effect of the number of objective values per batch on the controller by randomly selecting only a subset of the objectives $f(m, \theta)$ for the controller $p_\phi$ to use, but maintain the original number of workers for updating $\theta_s$ via ES to maintain weight estimation quality to prevent confounding results. We found that this sample reduction can reduce the performance of both controllers for various tasks, especially the PG controller. Thus, we find the **use of the already present ES workers highly crucial** for the controller's quality of architecture search in this setting.

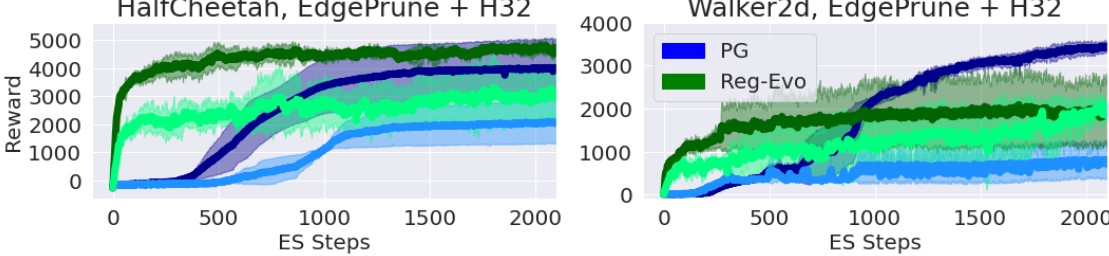

Figure 5: Regular ES-ENAS experiments with 150 full controller objective value usage plotted in darker colors. Experiments with lower controller sample usage (10 random samples, similar to the number of simultaneously training models in (Tan et al., 2018b)) plotted in corresponding lighter colors.

### 4.3.3 Constrained Optimization

Following (Tan and Le, 2019; Tan et al., 2018b) on similar techniques for *constrained optimization*, the controller may optimize multiple objectives (ex: efficiency) towards a Pareto optimal solution (Deb, 2005). We apply (Tan et al., 2018b) and modify the controller's objective to be a hybrid combination $f(m, \theta) \left( \frac{|E_m|}{|E_T|} \right)^\omega$ of both the total reward $f(m, \theta)$ and the compression ratio $\frac{|E_m|}{|E_T|}$ where $|E_m|$ is the number of edges in model $m$ and $|E_T|$ is a target number, with the search space expressed as boolean mask mappings $(i, j) \rightarrow \{0, 1\}$ over all possible edges. For simplicity, we use the naive

setting in (Tan et al., 2018b) and set $\omega = -1$ if $\frac{|E_m|}{|E_T|} > 1$, while $\omega = 0$ otherwise, which strongly penalizes the controller if it proposes a model $m$ whose edge number $|E_m|$ breaks the threshold $|E_T|$.

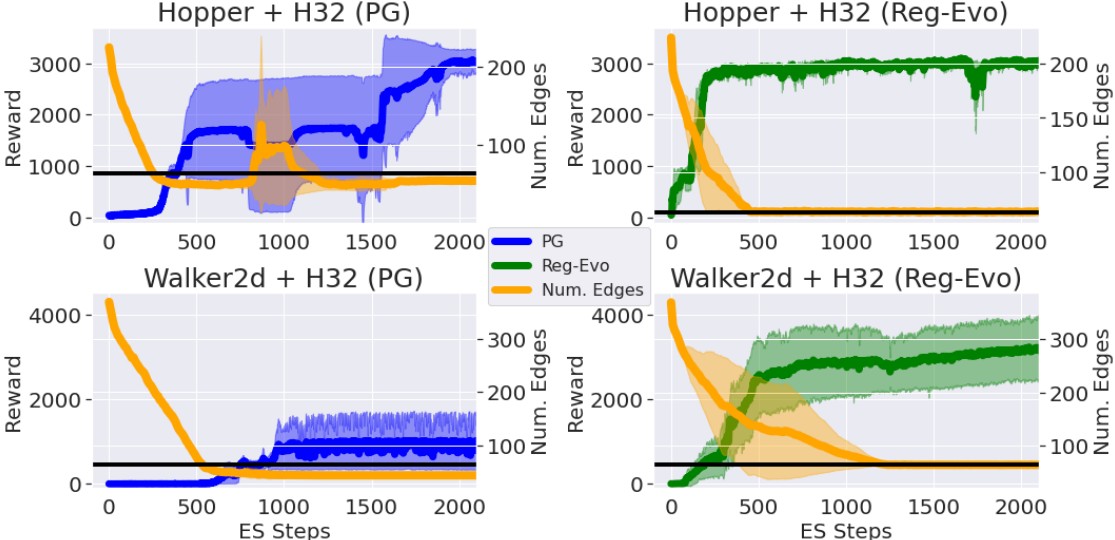

Figure 6: Environment reward plotted alongside the average number of edges used for proposed models. **Black** horizontal line corresponds to the target $|E_T| = 64$.

In Fig. 6, we see that the controller eventually reduces the number of edges below the target threshold set at $|E_T| = 64$, while still maintaining competitive training reward, demonstrating that ES-ENAS is also capable of constrained optimization techniques, potentially useful for explicitly designing efficient CPU-constrained robot policies (Unitree, 2017; Gao et al., 2020; Tan et al., 2018a).

## 5 Conclusions, Limitations, and Broader Impact Statement

**Conclusion:** We presented a scalable and flexible algorithm, ES-ENAS, for performing optimization over large hybrid spaces. ES-ENAS is efficient, simple, and modular, and can utilize many techniques from both the continuous and combinatorial evolutionary literature.

**Limitations:** In certain scenarios where $m$ specifies a model and thus the continuous parameter size $d$ is dependent on $m$, there may not be an obvious way to form a global $\theta$. This is a common issue that usually requires domain-specific knowledge (e.g. NAS) to resolve. Furthermore, due to reasons of simplicity, the joint sampling distribution $P_{m,\theta}$ over $(\mathcal{M}, \theta)$ was made as a product between independent distributions over $\mathcal{M}$ and $\theta$ in this paper. However, it may be worth studying distributions and update rules in which $m$ and $\theta$ are sampled dependently, as it may lead to even more effective algorithms.

**Broader Impact:** We believe that many large-scale evolutionary projects once prohibited by the curse of continuous dimensionality may now be feasible by the efficiency of ES-ENAS, potentially reducing computation costs dramatically. For example, one may be able to extend (Real et al., 2020) to also search for continuous parameters (e.g. neural network weights) via ES-ENAS. Furthermore, ES-ENAS is applicable to several downstream applications, such as architecture design for mobile robotics, and recently new ideas in RNNs for meta-learning and memory (Bakker, 2001; Najarro and Risi, 2020). ES-ENAS can potentially also be used for more broad scenarios involving evolutionary

search, such as genetic programming (Co-Reyes et al., 2021), circuit design (Ali et al., 2004), and compiler optimization (Cooper et al., 1999). Other potential applications include flight optimization (Ahmad and Thomas, 2013), protein and chemical design (Elton et al., 2019; Zhou et al., 2017; Yang et al., 2019), and program synthesis (Summers, 1977).

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

# Appendix

## A   Implementation Details

### A.1   API

We use the standardized NAS API PyGlove (Peng et al., 2020), where search spaces are usually constructed via combinations of *primitives* such as "`pyglove.oneof`" and "`pyglove.manyof`" operations, which respectively choose one item, or a combination of multiple objects from a container. These primitives can be combined in a nested conditional structure via "`pyglove.List`" or "`pyglove.Dict`". The search space can then be sent to an algorithm, which proposes child model instances $m$ programmically represented via Python dictionaries and strings. These are sent over a distributed communication channel to a worker alongside the perturbation $\theta + \sigma\mathbf{g}$, and then materialized later by the worker into an actual object such as a neural network. Although the controller needs to output hundreds of model suggestions, it can be parallelized to run quickly by multithreading (for Reg-Evo) or by simply using a GPU (for policy gradient).

### A.2   Algorithms

#### A.2.1   Combinatorial Algorithms

The mutator used for all evolutionary algorithms (Regularized Evolution, NEAT, Gradientless Descent/Batch Hill-Climbing) consists of a "Uniform" mutator for the neural network setting, where a parameter in a (potentially nested) search space is chosen uniformly at random, with its new value also mutated uniformly over all possible choices. For continuous settings, see Appendix A.3 below.

**Regularized Evolution:**   We set the tournament size to be $\sqrt{n}$ where $n$ is the number of workers/population size, as this works best as a guideline (Real et al., 2018).

**NEAT:**   We use the original algorithm specification of NEAT (Stanley and Miikkulainen, 2002) without additional modifications. The compatibility distance function was implemented appropriately for DNAs (i.e. "genomes") in PyGlove, and a gridsweep was used to find the best coefficients.

**Gradientless Descent/Batch Hill-Climbing:**   We use the same mutator throughout the optimization process, similar to (Song et al., 2020b) to reduce algorithm complexity, as the step size annealing schedule found in (Golovin et al., 2020) is specific to convex objectives only.

**Policy Gradient:**   We use a gradient update batch size of 64 to the Pointer Network, while using PPO as the policy gradient algorithm, with its default (recommended) hyperparameters from (Peng et al., 2020). These include a softmax temperature of 1.0, 100 hidden state size with 1 layer for the RNN, importance weight clipping of 0.2, and 10 update steps per weight update, with more values found in (Vinyals et al., 2015). We grid searched PPO's learning rate across $\{1 \times 10^{-4}, 5 \times 10^{-4}, 1 \times 10^{-3}, 5 \times 10^{-3}\}$ and found $5 \times 10^{-4}$ was the best.

#### A.2.2   Continuous Algorithms

**ARS/ES:**   We always use reward normalization and state normalization (for RL benchmarks) from (Mania et al., 2018b). For BBOB functions, we use $\eta_w = 0.5$ while $\sigma = 0.5$, along with 64

Gaussian directions per batch in an ES iteration, with 8 used for evaluation. For RL benchmarks, we use $\eta_w = 0.01$ and $\sigma = 0.1$, along with 75 Gaussian directions, with 50 more used for evaluation.

**CMA-ES:** For BBOB functions, we use $\sigma = 0.5$ and $\eta_w = 0.5$, similar to ARS/ES.

## A.3 BBOB Benchmarks

Our BBOB functions consisted of the 19 classical functions from (Hansen et al., 2009): {Sphere, Rastrigin, BuecheRastrigin, LinearSlope, AttractiveSector, StepEllipsoidal, Rosenbrock-Rotated, Discus, BentCigar, SharpRidge, DifferentPowers, Weierstrass, SchaffersF7, SchaffersF7IllConditioned, GriewankRosenbrock, Schwefel, Katsuura, Lunacek, Gallagher101}.

The each parameter in the raw continuous input space is bounded within $[-L, L]$ where $L = 5$. For discretization + categorization into a grid, we use a granularity of 1 between consecutive points, i.e. a categorical a parameter is allowed to select within $\{-L, -L+1, ..., 0, ..., L-1, L\}$. Note that each BBOB function is set to have its global optimum at the zero-point, and thus our hybrid spaces contain the global optimum.

Because each BBOB function may have a completely different scaling (e.g. for a fixed dimension, the average output for Sphere may be within the order of $10^2$ but the average output for BentCigar may be within $10^{10}$), we thus normalize the output of each function when reporting results. The normalized valuation of a BBOB function $f$ is calculated by dividing the raw value by the maximum absolute value obtained by random search.

Since for the ES component we use a step size of $\eta_w = 0.5$ and precision parameter of $\sigma = 0.5$, we thus use for evolutionary mutations, a Gaussian perturbation scaling $\sigma_{mut}$ of 0.07, which equalizes the average norms between the update directions on $\theta$, which are: $\eta_w \nabla_\theta \tilde{f}_\sigma$ and $\sigma_{mut}\mathbf{g}$.

## A.4 RL + Neural Network Setting

In order to allow combinatorial flexibility, our neural network consists of vertices/values $V = \{v_1, ..., v_k\}$, where the initial block of $|\mathcal{S}|$ values $\{v_1, ..., v_{|\mathcal{S}|}\}$ corresponds to the environment state, and the last block of $|\mathcal{A}|$ values $\{v_{k-|\mathcal{A}|+1}, ..., v_k\}$ corresponds to the action output values. Directed edges $E \subseteq E_{max} = \{e_{i,j} = (i, j) \mid 1 \leq i < j \leq k, |\mathcal{S}| < j\}$ are constructed with corresponding weights $W = \{w_{i,j} \mid (i, j) \in E\}$, and nonlinearities $G = \{\sigma_{|\mathcal{S}|+1}, ..., \sigma_k\}$ for the non-state vertices. Thus a forward propagation consists of for-looping in order $j \in \{|\mathcal{S}| + 1, ..., k\}$ and computing output values $v_j = \sigma_j \left( \sum_{(i,j) \in E} v_i w_{i,j} \right)$.

By default, unless specified, we use Tanh non-linearities with 32 units for each hidden layer.

**Edge pruning:** We group all possible edges $(i, j)$ into a set in the neural network, and select a fixed number of edges from this set. We can also further search across potentially different nonlinearities, e.g. $f_i \in \{\tanh, \text{sigmoid}, \sin, ...\}$ similarly to Weight Agnostic Neural Networks (Gaier and Ha, 2019). In terms of API, this search space can be described as `pyglove.manyof`$(E_{max}, |E|)$ along with `pyglove.oneof`$(\sigma_i, \mathcal{G})$. The search space is of size $\binom{|E_{max}|}{|E|}$ or $2^{|E_{max}|}$ when using a fixed or variable size $|E|$ respectively.

We collect all possible edges from a normal neural network into a pool $E_{max}$ and set $|E| = 64$ as the number of distinct choices, passed to the `pyglove.manyof`. Similar to quantization, this choice is based on the value $\max(|\mathcal{S}|, H)$ or $\max(|\mathcal{A}|, H)$, where $H = 32$ is the number of hidden units, which is linear in proportion to respectively, the maximum number of weights $|\mathcal{S}| \cdot H$ or $|\mathcal{A}| \cdot H$.

Since a hidden layer neural network has two weight matrices due to the hidden layer connecting to both the state and actions, we thus have ideally a maximum of $32 + 32 = 64$ edges.

For nonlinearity search, we use the same functions found in (Gaier and Ha, 2019). These are: {Tanh, ReLU, Exp, Identity, Sin, Sigmoid, Absolute Value, Cosine, Square, Reciprocal, Step Function.}

**Quantization:** We assign to each edge $(i, j)$ one color of many *colors* $c \in \mathcal{C} = \{1, ..., |\mathcal{C}|\}$, denoting the partition group the edge is assigned to, which defines the value $w_{i,j} \leftarrow w^{(c)}$. This is shown pictorially in Figs. 3a and 3b. This can also programmically be done by concatenating primitives `pyglove.oneof`($e_{i,j}$, $\mathcal{C}$) over all edges $e_{i,j} \in E_{max}$. The search space is of size $|C|^{|E|}$.

The number of partitions (or "colors") is set to $\max(|\mathcal{S}|, |\mathcal{A}|)$. This is both in order to ensure a linear number of trainable parameters compared to the quadratic number for unstructured networks, as well as allow sufficient parameterization to deal with the entire state/action values.

### A.4.1 Environment

For all environments, we set the horizon $T = 1000$. We also use the reward without alive bonuses for weight training as commonly used (Mania et al., 2018a) to avoid local maximum behaviors (such as an agent simply standing still to collect a total of 1000 reward), but report the final score as the real reward with the alive bonus.

### A.4.2 Baseline Details

We consider Unstructured, Toeplitz, Circulant and a masking mechanism (Choromanski et al., 2018; Lenc et al., 2019). We introduce their details below. Notice that all baseline networks share the same general (1-hidden layer, Tanh nonlinearity) architecture from A.4. This impplies that we only have two weight matrices $W_1 \in \mathbb{R}^{|\mathcal{S}| \times h}, W_2 \in \mathbb{R}^{h \times |\mathcal{A}|}$ and two bias vectors $b_1 \in \mathbb{R}^h, b_2 \in \mathbb{R}^{|\mathcal{A}|}$, where $|\mathcal{S}|, |\mathcal{A}|$ are dimensions of state/action spaces. These networks differ in how they parameterize the weight matrices. We have:

**Unstructured:** A fully-connected layer with unstructured weight matrix $W \in \mathbb{R}^{a \times b}$ has a total of $ab$ independent parameters.

**Toeplitz:** A toeplitz weight matrix $W \in \mathbb{R}^{a \times b}$ has a total of $a + b - 1$ independent parameters. This architecture has been shown to be effective in generating good performance on benchmark tasks yet compressing parameters (Choromanski et al., 2018).

**Circulant:** A circulant weight matrix $W \in \mathbb{R}^{a \times b}$ is defined for square matrices $a = b$. We generalize this definition by considering a square matrix of size $n \times n$ where $n = \max\{a, b\}$ and then do a proper truncation. This produces $n$ independent parameters.

**Masking:** One additional technique for reducing the number of independent parameters in a weight matrix is to mask out redundant parameters (Lenc et al., 2019). This slightly differs from the other aforementioned architectures since these other architectures allow for parameter sharing while the masking mechanism carries out pruning. To be concrete, we consider a fully-connected matrix $W \in \mathbb{R}^{a \times b}$ with $ab$ independent parameters. We also setup another mask weight $\Gamma \in \mathbb{R}^{a \times b}$. Then the mask is generated via

$$\Gamma' = \text{softmax}(\Gamma/\alpha)$$

where softmax is applied elementwise and $\alpha$ is a constant. We set $\alpha = 0.01$ so that the softmax is effectively a thresolding function wich outputs near binary masks. We then treat the entire concatenated parameter $\theta = [W, \Gamma]$ as trainable parameters and optimize both using ES methods. Note that this softmax method can also be seen as an instance of the continuous relaxation method from DARTS (Liu et al., 2019b). At convergence, the effective number of parameter is $ab \cdot \lambda$ where $\lambda$ is the proportion of $\Gamma'$ components that are non-zero. During optimization, we implement a simple heuristics that encourage sparse network: while maximizing the true environment return $f(\theta) = \sum_{t=1}^{T} r_t$, we also maximize the ratio $1 - \lambda$ of mask entries that are zero. The ultimate ES objective is: $f'(\theta) = \beta \cdot f(\theta) + (1 - \beta) \cdot (1 - \lambda)$, where $\beta \in [0, 1]$ is a combination coefficient which we anneal as training progresses. We also properly normalize $f(\theta)$ and $(1 - \lambda)$ before the linear combination to ensure that the procedure is not sensitive to reward scaling.

## B    Extended BBOB Experimental Results

### B.1    CMA-ES Comparison

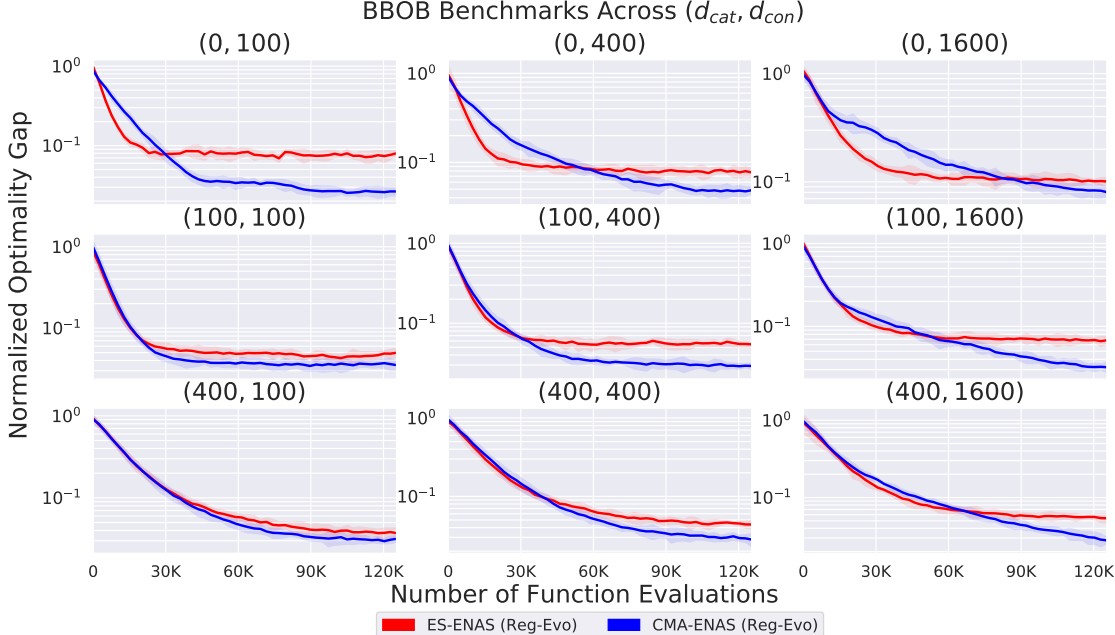

Figure 7: Comparison when regular ES/ARS is used as the continuous algorithm in ES-ENAS, vs when CMA-ES is used as the continuous algorithm (which we name "CMA-ENAS"). We use the exact same setting as Figure 2 in the main body of the paper. We use Regularized Evolution (Reg-Evo) as the default combinatorial algorithm due its strong performance found from Figure 2. We find that ES-ENAS usually converges faster initially, while CMA-ENAS achieves a better asymptotic performance. This is aligned with the results (in the first row) when comparing vanilla ES with vanilla CMA-ES. For generally faster convergence to a sufficient threshold however, ES/ES-ENAS usually suffices.

## C Extended Neural Network Experimental Results

As standard in RL, we take the mean and standard deviation of the final rewards across 3 seeds for every setting. "L", "H" and "H, H" stand for: linear policy, policy with one hidden layer, and policy with two such hidden layers respectively.

### C.1 Baseline Method Comparisons

In terms of the masking baseline, while (Lenc et al., 2019) fixes the sparsity of the mask, we instead initialize the sparsity at 50% and increasingly reward smaller networks (measured by the size of the mask $|m|$) during optimization to show the effect of pruning. Using this approach on several Open AI Gym tasks, we demonstrate that masking mechanism is capable of producing compact effective policies up to a high level of pruning. At the same time, we show significant decrease of performance at the 80-90% compression level, quantifying accurately its limits for RL tasks (see: Fig. 8).

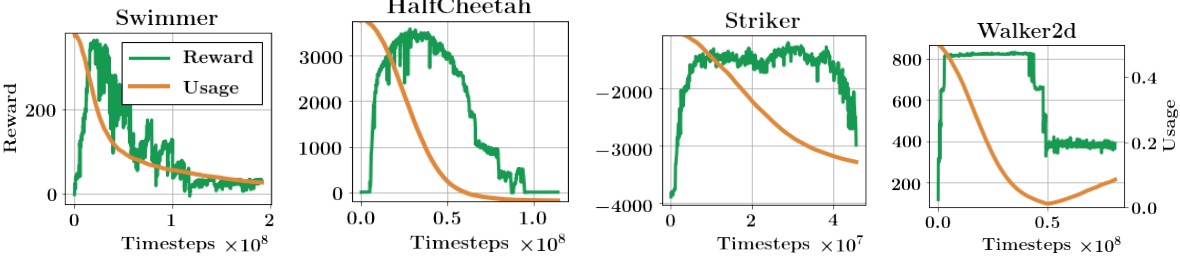

Figure 8: The results from training both a mask $m$ and weights $\theta$ of a neural network with two hidden layers. 'Usage' stands for number of edges used after filtering defined by the mask. At the beginning, the mask is initialized such that $|m|$ is equal to 50% of the total number of parameters in the network.

### C.2 Quantization

| Env. | Dim. | Arch. | Partitions | Policy Gradient | Regularized Evolution | Random Search |
|---|---|---|---|---|---|---|
| Swimmer | (8,2) | L | 8 | $366 \pm 0$ | $296 \pm 31$ | $5 \pm 1$ |
| Reacher | (11,2) | L | 11 | $-10 \pm 4$ | $-157 \pm 62$ | $-135 \pm 10$ |
| Hopper | (11,3) | L | 11 | $2097 \pm 788$ | $1650 \pm 320$ | $16 \pm 0$ |
| HalfCheetah | (17,6) | L | 17 | $2958 \pm 73$ | $3477 \pm 964$ | $129 \pm 183$ |
| Walker2d | (17,6) | L | 17 | $326 \pm 86$ | $2079 \pm 1085$ | $8 \pm 0$ |
| Pusher | (23,7) | L | 23 | $-68 \pm 2$ | $-198 \pm 76$ | $-503 \pm 4$ |
| Striker | (23,7) | L | 23 | $-247 \pm 11$ | $-376 \pm 149$ | $-590 \pm 18)$ |
| Thrower | (23,7) | L | 23 | $-819 \pm 8$ | $-1555 \pm 427$ | $-12490 \pm 708)$ |

| Env. | Dim. | Arch. | Partitions | Policy Gradient | Regularized Evolution | Random Search |
|---|---|---|---|---|---|---|
| Swimmer | (8,2) | H | 8 | $361 \pm 4$ | $362 \pm 1$ | $15 \pm 0$ |
| Reacher | (11,2) | H | 11 | $-6 \pm 0$ | $-23 \pm 11$ | $-157 \pm 2$ |
| Hopper | (11,3) | H | 11 | $3288 \pm 119$ | $2834 \pm 75$ | $95 \pm 2$ |
| HalfCheetah | (17,6) | H | 17 | $4258 \pm 1034$ | $4894 \pm 110$ | $-41 \pm 5$ |
| Walker2d | (17,6) | H | 17 | $1684 \pm 1008$ | $2026 \pm 46$ | $-5 \pm 1$ |
| Pusher | (23,7) | H | 23 | $-225 \pm 131$ | $-350 \pm 236$ | $-1049 \pm 40$ |
| Striker | (23,7) | H | 23 | $-992 \pm 2$ | $-466 \pm 238$ | $-1009 \pm 1$ |
| Thrower | (23,7) | H | 23 | $-1873 \pm 690$ | $-818 \pm 363$ | $-12847 \pm 172$ |

Table 2: Results via quantization across PG, Reg-Evo, and random search controllers. The number of partitions is always set to be $\max(|\mathcal{S}|, |\mathcal{A}|)$.

### C.3 Edge Pruning and Nonlinearity Search

Below in Table 3, we provide full results on edge-pruning.

| Env. | Dim. | Arch. | Policy Gradient | Regularized Evolution | Random Search |
|------|------|-------|-----------------|-----------------------|---------------|
| Swimmer | (8,2) | H | $105 \pm 116$ | $343 \pm 2$ | $21 \pm 1$ |
| Reacher | (11,2) | H | $-16 \pm 5$ | $-52 \pm 5$ | $-160 \pm 2$ |
| Hopper | (11,3) | H | $3349 \pm 206$ | $2589 \pm 106$ | $66 \pm 0$ |
| HalfCheetah | (17,6) | H | $2372 \pm 820$ | $4016 \pm 726$ | $-156 \pm 22$ |
| Walker2d | (17,6) | H | $3813 \pm 128$ | $1847 \pm 710$ | $0 \pm 2$ |
| Pusher | (23,7) | H | $-133 \pm 31$ | $-156 \pm 17$ | $-503 \pm 15$ |
| Striker | (23,7) | H | $-178 \pm 54$ | $-130 \pm 16$ | $-464 \pm 13$ |
| Thrower | (23,7) | H | $-532 \pm 29$ | $-1107 \pm 158$ | $-7797 \pm 112$ |

Table 3: Results via quantization across PG, Reg-Evo, and random search controllers. The number of edges is always set to be 64 in total, or (32, 32) across the two weight matrices when using a single hidden layer.

**Nonlinearity Search**   Intriguingly, we found that appending the extra nonlinearity selection into the edge-pruning search space improved performance across HalfCheetah and Swimmer, but not across all environments. However, lack of total improvement is consistent with the results found with WANNs (Gaier and Ha, 2019), which also showed that trained WANNs' performances matched with vanilla policies. From these two observations, we hypothesize that perhaps nonlinearity choice for simple MLP policies trained via ES are not quite so important to performance as other components, but more ablation studies must be conducted. Furthermore, for quantization policies, we see that hidden layer policies near-universally outperform linear policies, even when using the same number of distinct weights.

| Env. | Dim. | Arch. | Policy Gradient | Regularized Evolution | Random Search |
|------|------|-------|-----------------|-----------------------|---------------|
| Swimmer | (8,2) | H | $247 \pm 110$ | $359 \pm 5$ | $11 \pm 3$ |
| Hopper | (11,3) | H | $2270 \pm 1464$ | $2834 \pm 120$ | $57 \pm 7$ |
| HalfCheetah | (17,6) | H | $3028 \pm 469$ | $5436 \pm 978$ | $-268 \pm 29$ |
| Walker2d | (17,6) | H | $1057 \pm 413$ | $2006 \pm 248$ | $0 \pm 1$ |

Table 4: Results using the same setup as Table 3, but allowing nonlinearity search.

| Env. | Dim. | (PG, Reg-Evo) Reward | Method |
|------|------|----------------------|--------|
| HalfCheetah | (17,6) | $(2958, 3477) \rightarrow (4258, 4894)$ | Quantization (L $\rightarrow$ H) |
| Hopper | (11,3) | $(2097, 1650) \rightarrow (3288, 2834)$ | Quantization (L $\rightarrow$ H) |
| HalfCheetah | (17,6) | $(2372, 4016) \rightarrow (3028, 5436)$ | Edge Pruning (H) $\rightarrow$ (+ Nonlinearity Search) |
| Swimmer | (8,2) | $(105, 343) \rightarrow (247, 359)$ | Edge Pruning (H) $\rightarrow$ (+ Nonlinearity Search) |

Table 5: Rewards for selected environments and methods, each result averaged over 3 seeds. Arrow denotes modification or addition (+).

# D Network Visualizations

## D.1 Quantization

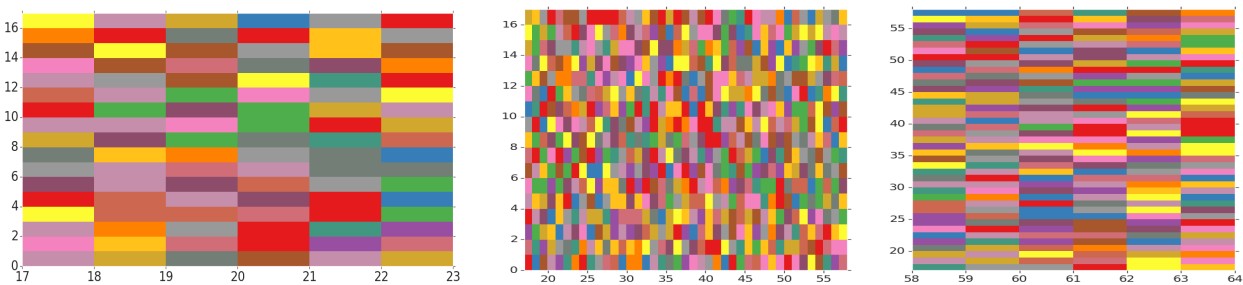

Figure 9: (a): Partitioning of edges into distinct weight classes obtained for the linear policy for HalfCheetah environment from OpenAI Gym. (b): Partitioning of edges for a policy with one hidden layer encoded by two matrices. State and action dimensionalities are: $s = 17$ and $a = 6$ respectively and hidden layer for the architecture from (b) is of size 41. Thus the size of the matrices are: $17 \times 6$ for the linear policy from (a) and: $17 \times 41$, $41 \times 6$ for the nonlinear one from (b).

## D.2 Visualizing and Verifying Convergence

We also graphically plot aggregate statistics over the controller samples to confirm ES-ENAS's convergence. We choose the smallest environment, Swimmer, which conveniently works particularly well with *linear* policies (Mania et al., 2018a), to reduce visual complexity and avoid permutation invariances. We also use a boolean mask space over all possible edges (search space size $|\mathcal{M}| = 2^{|\mathcal{S}| \times |\mathcal{A}|} = 2^{8 \times 2}$). We remarkably observe that *for all 3 independently seeded runs*, PG converges toward a "local maximum" architecture, demonstrated in Fig. 10 with the final architectures also presented for both PG and Reg-Evo. This suggests that there may be a few "natural architectures" optimal to the state representation.

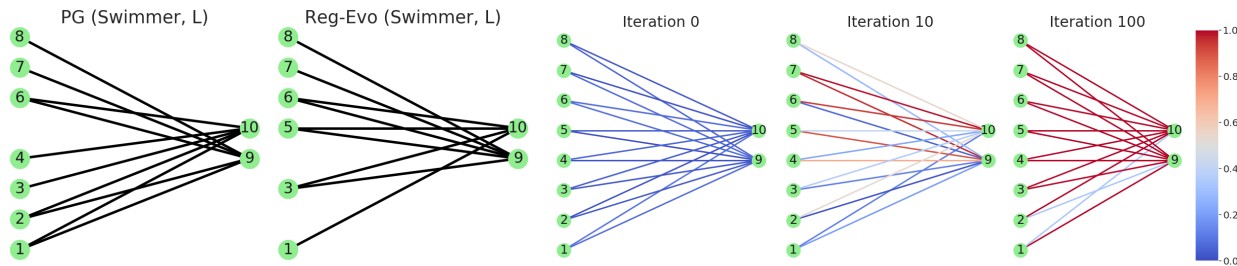

Figure 10: **Left:** Final architectures that PG and Reg-Evo converged to on Swimmer with a linear (L) policy, from the above runs. Note that the controller does not select all edges even if it is allowed in the boolean search space, but also *ignores some state values.* **Right:** Edge pruning convergence over time, with samples aggregated over 3 seeds from ES-ENAS using the PG controller on Swimmer. Each edge is colored according to a spectrum, with its color value equal to $2|p - \frac{1}{2}|$ where $p$ is the edge frequency. We see that initially, each edge has uniform ($p = \frac{1}{2}$) probability of being selected, but as both controller progress, their samples converge toward a single pruning.

## E    Theory

In this section, for convenience we use the variable $x$, which may be assigned $x = \theta$ in the main section of the paper. We present the ES/ARS and Mutation-based updates, which are respectively (assuming equal batch size $B$ of parallel workers):

$$x^+ = x + \eta \widehat{\nabla} \widetilde{f}_\sigma(x) \text{ where } \widehat{\nabla} \widetilde{f}_\sigma(x) = \sum_{i=1}^{B/2} \frac{f(x + \sigma \mathbf{g}_i) - f(x - \sigma \mathbf{g}_i)}{2\sigma} \mathbf{g}_i \tag{5}$$

$$x^+ = \arg\max\{f(x), f(x + \sigma_{mut}\mathbf{g}_1), ..., f(x + \sigma_{mut}\mathbf{g}_B)\} \tag{6}$$

We assume that $f$ is $\alpha$-strongly concave and $\beta$-smooth for $\alpha, \beta \geq 0$ if for all $x, y$:

$$\langle \nabla f(x), y - x \rangle - \frac{\beta}{2} \|y - x\|_2^2 \leq f(y) - f(x) \leq \langle \nabla f(x), y - x \rangle - \frac{\alpha}{2} \|y - x\|_2^2 \tag{7}$$

### E.1    ES/ARS Guarantees

We note that the $\beta$-smoothness also carries from the original function $f(x)$ into the smoothed function $\widetilde{f}_\sigma(x) = \mathbb{E}_{\mathbf{g} \sim \mathcal{N}(0,I)}[f(x + \sigma \mathbf{g})]$, and thus by simply combining the $\beta$-smoothness from Eq. 7 with the definition of $x^+$ from Eq. 5, we have

$$\eta \langle \nabla \widetilde{f}_\sigma(x), \widehat{\nabla} \widetilde{f}_\sigma(x) \rangle - \frac{\beta \eta^2}{2} \|\widehat{\nabla} \widetilde{f}_\sigma(x)\|_2^2 \leq \widetilde{f}_\sigma(x^+) - \widetilde{f}_\sigma(x) \tag{8}$$

Taking the expectation with respect to the sampling of $\mathbf{g}_1, ..., \mathbf{g}_{B/2}$ and noting that $\widehat{\nabla} \widetilde{f}_\sigma(x)$ is an unbiased estimation of $\nabla \widetilde{f}_\sigma(x)$:

$$\eta \|\nabla \widetilde{f}_\sigma(x)\|_2^2 - \frac{\beta \eta^2}{2} \left( \|\nabla \widetilde{f}_\sigma(x)\|_2^2 + \text{MSE}(\widehat{\nabla} \widetilde{f}_\sigma(x)) \right) \leq \Delta_{\sigma, ES}(x) \tag{9}$$

where $\Delta_{\sigma, ES}(x) = \mathbb{E}_{\mathbf{g}_1, ..., \mathbf{g}_{B/2} \sim \mathcal{N}(0,I)}[\widetilde{f}_\sigma(x^+)] - \widetilde{f}_\sigma(x)$ is the expected one-step improvement on the smoothed function $\widetilde{f}_\sigma$.

Using (Nesterov and Spokoiny, 2017), Theorem 4 leads to estimator variance $\text{MSE}(\widehat{\nabla} \widetilde{f}_\sigma(x)) = O(\beta^2 d^3 \sigma^2 / B)$ while Theorem 1 leads to $|f(x) - \widetilde{f}_\sigma(x)| = O(\sigma^2 \beta d)$, and finally Lemma 4 leads to $\|\nabla \widetilde{f}_\sigma(x)\|_2^2 - \|\nabla f(x)\|_2^2 = O(\beta^2 d^2 \sigma^2)$. Note that all of these terms are negligible compared to $\|\nabla f(x)\|_2^2$ as $\sigma$ is small and $B$ can be e.g. $O(d)$, and thus we may substitute these terms with single variables for the reader's convenience. Thus, this leads to:

$$G_0 + \eta(\|\nabla f(x)\|_2^2 + G_1) - \frac{\beta \eta^2}{2}(\|\nabla f(x)\|_2^2 + G_2) \leq \Delta_{ES}(x) \tag{10}$$

where the negligible terms are: $G_0 = -O(\sigma^2 \beta d), G_1 = O(\beta^2 d^2 \sigma^2), G_2 = O(\beta^2 d^2 \sigma^2 + \beta^2 d^3 \sigma^2 / B)$ and $\Delta_{ES}(x) = \mathbb{E}_{\mathbf{g}_1, ..., \mathbf{g}_{B/2} \sim \mathcal{N}(0,I)}[f(x^+)] - f(x)$ is the expected one-step improvement on the original $f$.

We may set $\eta = \frac{1}{\beta} \frac{\|\nabla f(x)\|^2 + G_1}{\|\nabla f(x)\|^2 + G_2} \approx \frac{1}{\beta}$ to maximize the quadratic (in terms of $\eta$) in the LHS, which leads to

$$\Theta\left(\frac{\|\nabla f(x)\|_2^2}{\beta}\right) = G_0 + \frac{1}{2\beta}\frac{(\|\nabla f(x)\|_2^2 + G_1)^2}{(\|\nabla f(x)\|_2^2 + G_2)} \leq \Delta_{ES}(x) \tag{11}$$

or in other words,

$$\Delta_{ES}(x) = \Omega\left(\frac{\|f(x)\|_2^2}{\beta}\right) \tag{12}$$

### E.2  Mutation Guarantees

We have from plugging in $y = x^+$ in Eq. 6 and 7 along with taking the expectation from sampling $\mathbf{g}_1, ..., \mathbf{g}_B$ and taking the argmax $\mathbf{g}_{max}$ (which can potentially also be zero if there is no improvement),

$$\Delta_{MUT}(x) \leq$$
$$\max\left(0, \mathbb{E}_{\mathbf{g}_1,...,\mathbf{g}_B\sim\mathcal{N}(0,I)}\left[\langle\nabla f(x), \sigma_{mut}\mathbf{g}_{max}\rangle\right] - \mathbb{E}_{\mathbf{g}_1,...,\mathbf{g}_B\sim\mathcal{N}(0,I)}\left[\frac{\alpha}{2}\|\sigma_{mut}\mathbf{g}_{max}\|_2^2\right]\right) \tag{13}$$

where $\Delta_{MUT}(x) = \mathbb{E}_{\mathbf{g}_1,...,\mathbf{g}_B\sim\mathcal{N}(0,I)}[f(x^+)] - f(x)$ is the expected improvement for the mutation.

We focus on upper bounding the non-zero term in the maximum in the RHS. Note that choosing $\mathbf{g}_{max} \in \{\mathbf{g}_1, ..., \mathbf{g}_B\}$ from the argmax process only optimizes $f(x + \sigma_{mut}\mathbf{g})$ and not any other objective, and thus:

$$\begin{aligned}
&\mathbb{E}_{\mathbf{g}_1,...,\mathbf{g}_B\sim\mathcal{N}(0,I)}[\langle\nabla f(x), \sigma_{mut}\mathbf{g}_{max}\rangle] \\
&\leq \sigma_{mut}\mathbb{E}_{\mathbf{g}_1,...,\mathbf{g}_B\sim\mathcal{N}(0,I)}\left[\max_{\mathbf{g}_i}\langle\nabla f(x), \mathbf{g}_i\rangle\right] \\
&\leq \sigma_{mut}\|\nabla f(x)\|_2\sqrt{2\log(B)}
\end{aligned} \tag{14}$$

where the bottom inequality is a well known fact about sums of Gaussians. For the other term, we have:

$$\mathbb{E}_{\mathbf{g}_1,...,\mathbf{g}_B\sim\mathcal{N}(0,I)}\left[\frac{\alpha}{2}\|\sigma_{mut}\mathbf{g}_{max}\|_2^2\right] \geq \frac{\alpha\sigma_{mut}^2}{2}\mathbb{E}_{\mathbf{g}_1,...,\mathbf{g}_B\sim\mathcal{N}(0,I)}\left[\min_{\mathbf{g}_i}\|\mathbf{g}_i\|_2^2\right] \tag{15}$$

To bound the RHS's right side, we may use a well-known concentration inequality for Lipschitz functions with respect to Gaussian sampling, i.e. $\Pr_{\mathbf{g}\sim\mathcal{N}(0,I)}[|M(\mathbf{g}) - \mu| > \lambda] \leq 2e^{-\lambda^2/2}$ where $M(\cdot)$ is any Lipschitz function and $\mu = \mathbb{E}_{\mathbf{g}'\sim\mathcal{N}(0,I)}[M(\mathbf{g}')]$. We may define $M(\mathbf{g}) = \|\mathbf{g}\|_2$ which leads to $\mu = \sqrt{d}$, and then use a union bound over $B$ IID samples to obtain:

$$\begin{aligned}
\Pr_{\mathbf{g}_1,...,\mathbf{g}_B\sim\mathcal{N}(0,I)}\left[\|\mathbf{g}_i\|_2 \geq \sqrt{d} - \lambda, \;\; \forall\mathbf{g}_i\right] &\geq \Pr_{\mathbf{g}_1,...,\mathbf{g}_B\sim\mathcal{N}(0,I)}\left[|\|\mathbf{g}_i\|_2 - \sqrt{d}| \leq \lambda, \;\; \forall\mathbf{g}_i\right] \\
&\geq (1 - B \cdot 2e^{-\lambda^2/2})
\end{aligned} \tag{16}$$

This finally implies that from Eq. 15,

$$\mathbb{E}_{\mathbf{g}_1,\ldots,\mathbf{g}_B \sim \mathcal{N}(0,I)} \left[ \min_{\mathbf{g}_i} \|\mathbf{g}_i\|_2^2 \right] \geq \max\left(0, \sqrt{d} - \lambda\right)^2 \cdot \Pr_{\mathbf{g}_1,\ldots,\mathbf{g}_B \sim \mathcal{N}(0,I)} \left[ \|\mathbf{g}\|_2 \geq \sqrt{d} - \lambda, \quad \forall \mathbf{g}_i \right]$$

$$\geq \max\left(0, \sqrt{d} - \lambda\right)^2 \cdot (1 - B \cdot 2e^{-\lambda^2/2}) \tag{17}$$

To set the probability-like term $(1 - 2Be^{-\lambda^2/2})$ in the RHS to a constant $C$, we let $\lambda = \sqrt{2\log(\frac{2B}{1-C})} = \Theta\left(\sqrt{\log(B)}\right)$, which finally leads to

$$\mathbb{E}_{\mathbf{g}_1,\ldots,\mathbf{g}_B \sim \mathcal{N}(0,I)} \left[ \min_{\mathbf{g}_i} \|\mathbf{g}_i\|_2^2 \right] \geq \max\left(0, \Theta\left(\sqrt{d} - \sqrt{\log(B)}\right)\right)^2 \tag{18}$$

Thus replacing the two terms in Eq. 13,

$$\Delta_{MUT}(x) \leq \max\left(0, \sigma_{mut}\|\nabla f(x)\|_2\sqrt{2\log(B)} - \alpha\sigma_{mut}^2 \max\left(0, \Theta\left(\sqrt{d} - \sqrt{\log(B)}\right)\right)^2\right) \tag{19}$$

If $B = \Omega(\exp(d))$, then there is no quadratic in terms of $\sigma_{mut}$, and thus $\sigma_{mut}$ can be arbitrarily large (or maximized at the search space's bounds) to essentially brute force the entire search space. Otherwise, hyperparameter tuning for $\sigma_{mut}$ leads to maximizing the quadratic in the RHS, which leads to setting $\sigma_{mut} = \Theta\left(\frac{\|\nabla f(x)\|_2\sqrt{2\log(B)}}{\alpha\cdot\left(\sqrt{d} - \sqrt{\log(B)}\right)^2}\right)$, leading to

$$\Delta_{MUT}(x) = O\left(\frac{\|\nabla f(x)\|_2^2\log(B)}{\alpha\cdot\left(\sqrt{d} - \sqrt{\log(B)}\right)^2}\right) \tag{20}$$

### E.3 Putting things together

Putting the expected improvements together, we see that:

$$\Delta_{MUT}(x) = O\left(\frac{\|\nabla f(x)\|_2^2\log(B)}{\alpha\cdot\left(\sqrt{d} - \sqrt{\log(B)}\right)^2}\right) \tag{21}$$

$$\Delta_{ES}(x) = \Omega\left(\frac{\|\nabla f(x)\|_2^2}{\beta}\right) \tag{22}$$

and thus there is a expected improvement ratio bound when $B = o(\exp(d))$:

$$\frac{\Delta_{ES}(x)}{\Delta_{MUT}(x)} = \Omega\left(\frac{1}{\kappa}\frac{\left(\sqrt{d} - \sqrt{\log(B)}\right)^2}{\log(B)}\right) \tag{23}$$

where $\kappa = \beta/\alpha$ is the condition number.

