# OpenReview forum: "ES-ENAS: Efficient Evolutionary Optimization for Large-Scale Hybrid Search Spaces"
_TMLR — Rejected by TMLR_

### Review · Reviewer_M5Xg · 2022-11-16

**Summary Of Contributions:**

This work proposes ES-ENAS, an Evolutionary Strategies (ES) based Efficient Neural Architecture Search (ENAS) method for solving large-scale black-box optimization problems with hybrid (e.g., combinatorial and continuous) search space. The key idea, inspired by ENAS, is to build a model-based neural combinatorial controller to optimize the combinatorial variables (via policy gradient), which can be optimized jointly with the continuous variables (via ES). Experimental results show that the proposed ES-ENAS approach performs well on the modified BBOB black-box optimization problems and architecture search over the Mujoco RL benchmarks.


**Audience:**

Yes

**Broader Impact Concerns:**

N.A.

**Claims And Evidence:**

No

**Requested Changes:**

Please address the major concerns listed in the weaknesses, it is crucial to provide

1) A clear discussion on the contribution and difference over ENAS, in addition to the simple usage of ES.

2) A strong and solid analysis to support the main claim on ES-ENAS for general black-box optimization.

3) A better justification for using pointer network as the controller (optimizer) for general combinatorial parameters.

4) A more meaningful complexity analysis for ES-ENAS on hybrid search space.

5) Comprehensive experiments for ES-ENAS on general real-world black-box optimization problems rather than NAS (to support point 2).

**Strengths And Weaknesses:**

**Strengths**

- This work is generally well-organized and easy to follow.
- Large-scale black-box optimization problem, especially with hybrid search space, is important while challenging for many real-world applications. This work is a timely contribution to an important topic, which could be interesting for many TMLR audiences.
- The proposed ES-ENAS method is simple and can be easily implemented. However, there are also some major concerns on the proposed method, see the weaknesses below.

**Weaknesses**


**1. Contribution and Motivation**

The key ideas for ES-ENAS, such as parameter sharing, model-based neural combinatorial controller, and joint optimization with combinatorial/continuous parameters, are all directly from the seminal ENAS approach [1]. The only difference seems to be the usage of ES for black-box (non-differentiable) optimization on the continuous parameters \theta, rather than the gradient-based method in ENAS. However, as correctly mentioned in this work, ES has already been widely used for non-differentiable black-box optimization and reinforcement learning. The contribution of simply replacing the gradient-based method in ENAS with ES is weak.

This work claims the advantage of ES-ENAS over ENAS is for fully black-box optimization involving thousands of CPU-only workers. However, this benefit is solely from the simple usage of ES. The motivation for general black-box optimization is also not solid (see next point).


**2. ES-ENAS is NOT for General Black-Box Optimization**

As indicated by this work (e.g., the title, abstract, introduction, and conclusion), ES-ENAS is designed for general black-box optimization with large-scale hybrid search space. However, this claim is not well supported.

Like ENAS, ES-ENAS requires the optimization problem has global continuous parameters \theta that can be shared by different combinatorial structures m. This setting could be specific to the NAS problem where m is different sub-models from a single shared supernet with weight \theta, but it is very hard to see why it could be the case for a general black-box optimization problem. For a general problem, the (m, \theta) pair could be different for different solutions. Why ES-ENAS, as a direct extension from the ENAS algorithm specific for neural architecture search, could be suitable for general black-box optimization?

In the experiments, ES-ENAS is tested on a set of modified BBOB problems (not its standard form), and the RL NAS problem for Mujoco benchmarks. Comprehensive experiments on different real-world large-scale black-box optimization problems (rather than NAS) are needed to truly support the claim that ES-ENAS is a general black-box optimizer.


**3. Pointer Network as the Combinatorial Optimizer**

This work proposes to use a pointer network [2] as the controller (optimizer) for the combinatorial parameters. However, this choice is not well-supported, especially for general black-box optimization problems.

To my understanding, the pointer network is mainly designed for the specific combinatorial optimization problem with sequential output, where the size of the output dictionary depends on the input size and the valid output dictionary could change for each step. Therefore, it is suitable to solve the combinatorial problem with special structures such as convex hull,  Delaunay triangulation, traveling salesman problem, and other combinatorial problems with sequential outputs. However, for a general black-box optimization problem, the combinatorial parameters could indeed be the categorical or simple discrete variables but not with the structures mentioned above.

The Pointer network needs to model the combinatorial problem instance as an input sequence, which could be not suitable for a general optimization problem. Indeed, there is rich literature on the neural combinatorial optimization (NCO) approach [3,4,5] which has shown that the sequence-based encoder (as in the pointer network) is not ideal for different combinatorial optimization problems.

These NCO approaches are also typically optimized by policy gradient descent for combinatorial optimization. The proposed approach in this work (for updating the controller for a given problem) is equal to the active search approach without pre-training in NCO [3]. But again, these approaches are for combinatorial problems with specific structures but not a general black-box optimization problem.

**4. Complexity Analysis**

The main theoretical analysis result is to show ES only requires O(d) evaluations while the mutation-based method requires O(exp(d)) evaluations for **continuous optimization problem**. However, since the motivation of ES-ENAS is mainly for solving large-scale black-box optimization problems with hybrid search spaces, it is more interesting to know the complexity analysis for **combinatorial optimization problem**.

In addition, with the model-based optimizer, the dimension of optimization parameters could be significantly different. The simple mutation-based method will have a small d (e.g., 100) for the combinatorial parameters, while the controller will have a much larger D for model parameters (e.g., more than 100,000 for pointer network).

**Other Comments**

- In the abstract, there is a typo in "smoothed gradient gradient techniques".

- In ES-ENAS, the combinatorial controller (e.g., pointer network) is optimized with the policy gradient method but not the supervised learning method in the original pointer network paper. The work on pointer network with reinforcement learning (policy gradient) [3] should be cited.

**Reference**

[1] Efficient neural architecture search via parameter sharing. ICML 2018.

[2] Pointer networks. NeurIPS 2015.

[3] Neural Combinatorial Optimization with Reinforcement Learning. ICLR workshop 2017.

[4] Reinforcement learning for solving the vehicle routing problem. NeurIPS 2018.

[5] Attention, learn to solve routing problems! ICLR 2019.

---

> ### Author Response · Authors · 2022-11-20
> **Response to Reviewer M5Xg**
>
> We thank the reviewer for their very fast response! We are very grateful for the feedback, and have made relevant changes to the paper. We also provide explanations below.
>
> # Responses
> **1. Contribution and Motivation**
>   * **Contribution:** We definitely agree that the building blocks (ES and ENAS) are well established, but believe this is actually a strength highlighting the simplicity, ease-of-setup, and modularity of our method, allowing any blackbox optimizers to be swapped in.
>   * **Motivation:** We believe ES-ENAS fills a crucial gap in both the RL/Robotics and NAS communities, which is the study of **NAS on large non-differentiable networks/policies.** These class of networks include:
>     * For stable robotic control, any deterministic policies which cannot be trained using policy gradients (which require stochastic policies).
>     * Recent non-standard networks such as Hebbian networks [1] and even classics such as discrete-weight [2] and switching circuits [3].
>
> **2.  “ES-ENAS is NOT for General Black-Box Optimization”:** We definitely agree with this statement, but believe we did not state explicitly that ES-ENAS covers all blackbox search spaces in our paper, rather only for hybrid search spaces of the separable form $(\mathcal{M}, \mathbb{R}^{d})$.
>   * However, we definitely want to be careful with our wording and avoid any implications or misleadings to the full general case. We modified the wording of our paper in various places (highlighted in red) to be more precise.
>   * We also do note that many spaces, where $d$ depends on the combinatorial $m$, can indeed be reparameterized into our $(\mathcal{M}, \mathbb{R}^{d})$ formulation, by computing the union of all continuous variables over all $m \in \mathcal{M}$ (similar to the supernet formulation in NAS).
>
> **3. Pointer Networks:**
>   * **Our reason for use in paper:** We firstly want to emphasize that our intention is not to force the use of the Pointer Network controller, as even from our experiments, it often loses to other controllers such as Regularized Evolution. Rather, we used the Pointer Network as a popular and common baseline to emphasize the flexibility of the ES-ENAS framework to any combinatorial controller.
>   * **"Pointer Networks not applicable to general blackbox optimization (e.g. categorical/discrete variables)":** We respectfully disagree. For instance, the Policy Gradient / Pointer Network from PyGlove [4] supports basic blackbox optimization primitives such as the categorical `pg.oneof` and n-choose-k `pg.manyof` by imposing a sequential ordering on categorical/discrete variables.
>     * We have included the supplemental PPO optimizer code (provided privately from the PyGlove authors) for review to demonstrate how it supports basic categorical/discrete variables.
>
> **4. Complexity Analysis:**
>   * **Theoretical analysis for combinatorial case:** This generally infeasible, for various reasons (and hence the supporting BBOB empirical analysis):
>     * **Combinatorial search spaces can be of any structure** (e.g. flat categoricals, trees, graphs, and even program synthesis [5]) which makes a general theory over all such spaces impossible.
>     * **Lack of notion of “convexity”:** Convergence proofs generally require a notion of convexity of the objective to allow consistent improvement. Even for pure continuous spaces, this is commonly the case in order to provide a meaningful proof. For general combinatorial search spaces, this may be mathematically impossible or require very contrived additional assumptions that makes a resulting proof overly specific to said assumptions.
>     * **Lack of previous theory:** For example, even the very popular and relatively simple Regularized Evolution lacks theoretical analysis, which also would make the ES-ENAS variant proof impossible.
>   * **[Pointer Network] Controller having many parameters:** We believe this is a misconception and we’d like to clarify that the sample complexity is dependent and bottlenecked only by the *non-differentiable parameter space* (i.e. the original dimension $d$ of the blackbox function). For a differentiable controller, what matters is thus how fast the output distribution $p_{\phi}$ of the controller moves in the original space $\mathbb{R}^{d}$.
>     * The parameters $\phi$ on the differentiable controller are updated by auto-differentiation and do not suffer from sample complexity effects (analogous to being able to train very large neural network classifiers via auto-diff).
>
> # References
> [1] [Meta-Learning through Hebbian Plasticity in Random Networks](arxiv.org/pdf/2007.02686.pdf)
>
> [2] [A logical calculus of the ideas immanent in nervous activity](cs.cmu.edu/~./epxing/Class/10715/reading/McCulloch.and.Pitts.pdf)
>
> [3] [Adaptive switching circuits](www-isl.stanford.edu/~widrow/papers/c1960adaptiveswitching.pdf)
>
> [4] [PyGlove](github.com/google/pyglove)
>
> [5] [AutoML-Zero: Evolving Machine Learning Algorithms From Scratch](arxiv.org/abs/2003.03384)

---

> > ### Author Response · Authors · 2022-11-20
> > **Response to Reviewer M5Xg (Continued)**
> >
> > # Requested Changes:
> > Please see our answers above for any clarifications. For concrete actions, we have correspondingly **(changes highlighted in red in the paper)**:
> > 1. Discussed that ES-ENAS is crucial for search over large non-differentiable networks.
> > 2. Made our wording more careful to avoid any implications that our method covers all combinatorial spaces (in particular, spaces where $d$ depends on $m$ and cannot be reparameterized into our hybrid formulation).
> > 3. Included the code for the Policy Gradient / PPO optimizer from PyGlove in the supplementary materials to demonstrate its applicability to generic categorical/discrete variables.
> > 4. Wrote in an additional note about infeasibility of theoretical analysis.
> > 5. We believe we have clarified our scope on point 2, and believe that our BBOB experiments are sufficient to demonstrate use on the blackbox optimization case and ES-ENAS’s efficient sample complexity.
> >
> > Typos and Citations: Fixed and added, thanks.
> >
> > **Please let us know if you have any more questions or concerns, thank you!**

---

> > > ### Comment · Reviewer_M5Xg · 2022-12-13
> > > **Thank you for the detailed response**
> > >
> > > Thank you for the detailed responses, and some of my concerns have been properly addressed. Here are some follow-up comments:
> > >
> > > **1 Motivation & 2 General Solver**
> > >
> > > The proposed ES-ENAS is currently more like a specific NAS algorithm (with a black-box function), since it is a direct extension from ENAS and the main experimental study is also on NAS for RL.
> > >
> > > I do agree ES-ENAS can be applied to more general problems with the separate (M,R^d) formulation, and the various examples like chemical design and problem synthesis mentioned in this paper are also promising. However, without an experimental study on these problems (rather than NAS), it is not solid enough to claim the proposed method is for (a more general) large-scale hybrid search spaces.
> > >
> > > Therefore, the possible ways to address this concern is to:
> > >
> > >    - further modify the wording, and clearly state that this work is for NAS with a black-box function.
> > >
> > >    - or run experiments on other real-world applications rather than NAS with the (M,R^d) formulation to support the claim.

---

> > > > ### Author Response · Authors · 2022-12-15
> > > > **Response on Blackbox Functions**
> > > >
> > > > Thank you for the follow-up!
> > > >
> > > > From draft updates from the other reviewers since our last thread as well in addition to this thread, we have moved the mention of broad applications from the introduction to the future works and conclusion section.
> > > >
> > > > Please note that our Section 3.1, Figure 2 also discusses the performance of ES-ENAS on modified BBOB functions, which are very common benchmark tasks in the blackbox optimization literature, outside of NAS.

---

### Review · Reviewer_HAf2 · 2022-12-01

**Summary Of Contributions:**

This paper proposes a black-box optimization framework for mixed combinatorial / continuous search space. An example target problem is an architecture search for deep reinforcement learning.

The proposed framework is based on a parallel evolution strategy popularized by Salimans et al. (2017). To tackle the mixed combinatorial / continuous search space, a sampling distribution (usually normal distribution in case of a continuous search space) is replaced with the product of a distribution over combinatorial search space (called a controller) and a distribution over the continuous search space. The distribution mean for the continuous part is updated using the evolution strategy by estimating the gradient of the smoothed objective (expectation of the objective under the distribution over the mixed search space) by sampling random solutions. The update of the controller is abstracted, i.e., one can plug-in a possibly domain-specific approach. The resulting framework is named ES-ENAS.

The proposed approach is empirically evaluated on modified BBOB test problems with different baseline component for the controller optimizer to show the usefulness of simultaneously optimizing both continuous and combinatorial parameters rather than just optimizing combinatorial parameters. The proposed approach is then applied to a neural architecture search for reinforcement learning setting. Compared to the approaches using hand-designed pattern and optimization-based approaches, the proposed approach exhibits greater final rewards with smaller networks.


**Audience:**

Yes

**Broader Impact Concerns:**

no particular concerns

**Claims And Evidence:**

No

**Requested Changes:**

See the weaknesses section. These three points are required to address.

Below is the other corrections / concerns that should be addressed:

- P1: S \to A: Both S and A are not introduced previously.
- P3: "which be practically" -> "which is practically"
- P3: in Algorithm 1. p_\phi is defined as a distribution, but the set of two pairs are assigned to it.
- P4: Policy Gradient Methods: It looks like sigma is missing in front of g at two places.
- P6-7: I suggest to explain the setting in the main text. I personally think that it is hard to understand the setting of this experiments as most of the details are moved to Appendix. It was not clear how the number of colors are determined, and how the solution is encoded.
- P7: The authors say that the training have been performed until convergence. However, if the training time is not the same for different methods, the comparison looks unfair. Fast approaches can perform more restarts within a fixed budget and one should take the best results of the restarts done in the given budget.


**Strengths And Weaknesses:**

Strengths:

- Simplicity and modularity of the proposed approach. It is advantageous as a domain-specific combinatorial optimizer can be relatively easily plugged-in.

Weaknesses:

- Missing Baselines: The proposed approach has not been compared with other approaches that can be applied to the same problem. In Section 3, the proposed approaches using different combinatorial optimizers are compared to their combinatorial only counterparts. It reveals the importance of considering continuous variables and shows the proposed approach can do better than the combinatorial only counterparts. However, it has not been compared to existing baseline approaches. For example, ENAS (Pham et al, 2018) itself can be simply extended to treat continuous parameters by introducing the product of the normal distribution and the categorical distribution as the controller. Akimoto et al., (2019) indeed did this extension (Section 2.4): they consider the mix of continuous and combinatorial architecture variables that are derivative-free, as well as continuous weights of the supernet that are differentiable. I think that this baseline (removing the continuous weights part of course) is as general as the proposed approach except that the update of the distribution of the combinatorial parameter is not modularized. Moreover it is applicable to the experiments in Section 4 as well. To show the usefulness of the proposed approach over existing approaches, the authors should compare the proposed approach with these existing baseline approaches.

- Claims and Evidence 1: The author claims that they achieve significantly more sample efficiency in the abstract. However, no experimental evaluation in this paper reveals that the weight sharing one shot optimization achieves better sample efficiency. I couldn't find any discussion about the number of interaction steps in Section 4. Is this really more sample efficient than existing approaches? I think some existing approaches such as NEAT can be applied to the same problem setting (at least for the setting of pruning), possibly with some regularization for the network size. Because NEAT is not one-shot, the sample efficiency of the proposed approach must be better than NEAT, but it is not clear if this is true. It is also not clear whether this is really more sample efficient than a very naive baseline that trains a neural networks that is increased progressively like in Progressive Neural Networks (Rusu et al., 2016).

- Claims and Evidence 2: The author claims that the proposed approach does not have an issue of degradation of the performance with increasing dimension. However, it is definitely an wrong message. There is no reason that the problem is solved by the combination. At least, one can not derive it from Figure 2. What I see in Figure 2 is that if a problem has categorical variables, they needs to be optimized as well as continuous variables. It may be the sase that the categorical variables are easier to optimize in these test problems and hence with a fixed number of function evaluations the optimality gap can quickly drops. However, if one focus on the optimality gap due to the continuous variables, I suspect that the same issue, slow down of convergence, must appear.

- Statistical Significance: The experiments in Section 4  is conducted only 3 times and only the best results are reported. No statistical significance is shown. How large the standard deviations are for example? Moreover, because it is RL tasks, the cumulative reward itself is stochastic. How many episodes have been performed to compute the rewards?

---

> ### Author Response · Authors · 2022-12-05
> **Response to Reviewer HAf2**
>
> We thank the reviewer for their rich insights into baselines and acknowledgement of the simplicity + modularity of our method! In general, we would like to highlight the core merits to our ES-ENAS method, while also emphasizing that the goal of our paper is not to claim ES-ENAS will outperform numerous baselines, but rather its competitiveness and generality.
>
> We also respond directly below to raised points, along with an updated paper draft:
>
> **Missing Baselines:**
> * **Extending ENAS (Pham et al, 2018) to continuous parameters:** The original implementation based on Pointer Networks does not support this, and due to its complicated codebase (please see supplemental code), we wanted to remain accurate to its original implementation. While extending to continuous parameters is possible and a great point, such a result most likely would support our claim in Section 3, as this would be another gradient-based relaxation method on the continuous space (similar to ES-ENAS).
> * **Akimoto et al., (2019):** As we noted in Section 1, this baseline is not applicable as it only covers when $f$ is differentiable w.r.t continuous parameters (in their notation as “$x$”), which is usually specific to NAS over differentiable neural networks. The key difference is that our work covers when $f$ is fully blackbox, which is more broadly applicable to non-differentiable policies and applications where gradients do not exist.
>     * We quote from Section 2 of their paper: “Our objective is to simultaneously optimize $x$ and $c$ by possibly utilizing the gradient $\nabla_{x}f$”.
>
> **Claims and Evidence 1:**
>  * **Sample efficiency and NEAT as a baseline:** Our emphasis in this work is broadly optimization over any spaces of the form $(\mathcal{M}, \mathbb{R}^{d})$ as a whole, with the RL results in Section 4 as an example downstream application. We believe that the sample efficiency results in Section 3 have shown that NEAT (among other evolutionary methods such as Regularized Evolution) as a baseline can already suffer from high dimensional effects.
>  * **Progressive Neural Networks (Rusu et al., 2016) as a baseline:** We reiterate that our goal is to provide a method that can both be used for general-purpose blackbox optimization, with downstream applications to e.g. NAS in non-differentiable RL settings. Our intention is not to show that ES-ENAS is better than all other baselines, but at least performs competitively and can be used in applications such as the ones in Section 4.
>
> **Claims and Evidence 2:**
> * **Wording:** Much apologies for the wording, which we have fixed in Section 3.1; what we meant was that vanilla approaches will degrade much more severely than ES-ENAS over large continuous spaces.
> * **Continuous degradation:** In Figure 2, we have shown that for a given row (i.e. fixed categorical size), severe performance degradation occurs when $d_{con}$ (continuous size) increases, where vanilla methods fail to reduce the optimality gap, whereas ES/ES-ENAS based methods perform much better.
>
> **Statistical Significance:** We have added back the standard deviations of our results (originally omitted and moved to Appendix C to reduce space) with Toeplitz/Circulant baselines copied from relevant papers.
>
> **Other Corrections:** Done, thank you for the pointers!
>
>
> We would be happy to discuss in further detail if needed, thank you!

---

> > ### Comment · Reviewer_HAf2 · 2022-12-12
> > **Not convinced for missing baselines**
> >
> > Thank you for your reply. However, the comment on the missing baseline part is not convincing. As long as I understand, the approach by Akimoto et al. (2019) treats all of 1. weight parameters (differentiable), 2. categorical architecture parameter (black-box), and 3. continuous (or integer) architecture parameter (black-box), as I have written in my original review. Therefore, if one simply ignore the weight parameters there, one gets an algorithm that is applicable to the same problems that the proposed approach covers. In other words, I think the above method is more general as it covers wider class of problems.

---

> > > ### Author Response · Authors · 2022-12-13
> > > **On Akimoto et al. (2019)**
> > >
> > > Hi, we've taken a more careful examination of Akimoto et al. (2019); please feel free to correct us if our understanding is incorrect somewhere.
> > >
> > > We will use Akimoto et al. (2019)'s notation of denoting a function $f(x, \theta)$ where $x$ are differentiable weights and $\theta$ are parameters describing a distribution $p_{\theta}$ over a blackbox domain $\mathcal{C}$. Since $x$ is not applicable in our non-differentiable case, we will instead focus on $\theta$.
> > >
> > > We would like to clarify on the following points comparing our work to Akimoto et al. (2019):
> > >
> > > * **Generality:** While we agree that Akimoto et al. (2019) also describes the notion of a J-function $J(x, \theta) = \mathbb{E}_{c \sim p_\theta } [f(x, c)]$ and performs policy gradient updates $\nabla_\theta J(x, \theta)$,
> > >     * This formulation does not cover non-policy gradient updates, such as evolutionary algorithms like Regularized Evolution for the combinatorial space $\mathcal{C}$ nor CMA-ES for the continuous space (both of which we do cover in our work).
> > >     * Since we do not cover differentiable $x$, this leaves the focus on pure policy gradient methods over $\mathcal{C}$. In this case, we believe it is too broad to consider a work un-novel if it can be seen as a policy gradient, as implementation details matter very much, e.g. how to construct a distribution $p_{\theta}$ over $\mathcal{C}$ (Pointer Network vs Softmax relaxation vs Bernoulli Distribution), and how to update continuous non-differentiable parameters (ES vs VAE vs Correlations between Gaussians).
> > > * **Focus and Contribution:** We also believe that Akimoto et al. (2019)'s work focuses primarily on **(1)** Theoretical analysis of the interplay between $x$ and $\theta$, and **(2)** Application to autodiff-based NAS. Both are inapplicable in our setting, because we focus only on $\theta$ and non-differentiable architectures.
> > > * **Meaning of a baseline:** If our answers are not satisfactory, we respectfully would like to ask what it means to have Akimoto et al. (2019) as a baseline and what its purpose is. The experimental section of Akimoto. (2019) only considers distributionally parametrizing spaces which can be represented as flat search spaces consisting of only integer/categorical spaces.
> > >     * We note that while this is applicable to our "toy problems" in Section 3 regarding BBOB functions, it inapplicable to nested structures (our Section 4) and hence cannot considered (i.e. including/excluding it does not affect our contributions or main point).
> > >     * We would like to gently remind the reviewer that our focus is on optimizers which are applicable to highly nested and combinatorial structures, as they can be generally applicable to any combinatorial spaces.

---

> > > > ### Author Response · Authors · 2022-12-28
> > > > **Gentle Followup/Ping**
> > > >
> > > > Hi, and happy holidays! Please let us your thoughts on our response, if there is something we've missed or if our understanding is incorrect.
> > > >
> > > > We'd like to generally remind and emphasize that our intention is not to force the use of the Pointer Network controller, as even from our experiments, it can lose to other controllers such as Regularized Evolution. Rather, we used the Pointer Network as a popular and common method to emphasize the flexibility of the ES-ENAS framework to any combinatorial controller.
> > > >
> > > > Thanks again!

---

> > > > ### Comment · Reviewer_HAf2 · 2023-01-04
> > > > **Thanks for the clarification**
> > > >
> > > > and I am very sorry for my late response. Now I understand that the "generality" is for the combination of the search component for the categorical variables, not for the task. However, the usefulness of combining ES with some combinatorial optimizer rather than PG is not really demonstrated, i.e., the usefulness of the generality. In my opinion, the generality of the approach itself is not a sufficient contribution for the transaction, unless its usefulness is supported by experimental results.

---

> > > > > ### Author Response · Authors · 2023-01-04
> > > > > **Response about usefulness of generality**
> > > > >
> > > > > We are happy to discuss more on this topic, but we first need clarification on the phrase "Policy Gradient (PG)". Can we ask what the Reviewer means when they refer to "Policy Gradient" (in the blackbox optimization case)?
> > > > >
> > > > > * Is it referring to *any* method for estimating the gradient $\nabla_z J(z)$ of a smoothing $J(z) = \mathbb{E}_{x \sim p_z}[f(x)]$ where $x \in \mathcal{X}$ over any search space $\mathcal{X}$ and any blackbox function $f: \mathcal{X} \rightarrow \mathbb{R}$?
> > > > >   * In this case we agree that ES-ENAS (Pointer Network) variant essentially *is* a Policy Gradient method, since the ES subcomponent itself can be seen as a policy gradient, where one is estimating gradients over the smoothed function $J(\theta) = \mathbb{E}_{g \sim \mathcal{N}(0, I)}[f(\theta + \sigma g)]$.
> > > > >   * However, we feel that it is too harsh to say that our work does not contribute, since the term "Policy Gradient" is too broadly encompassing to nearly any local-search method which can be viewed as some form of gradient descent. An analogous argument would be to say that well-known algorithms such as [ARS](https://arxiv.org/pdf/1803.07055.pdf)/[ES](https://arxiv.org/abs/1703.03864) or even [CMA-ES](https://en.wikipedia.org/wiki/CMA-ES) are not sufficient or novel contributions simply because they can be seen as "just Policy Gradients".
> > > > >    * Furthermore, even the above does not convince the reviewer of our contributions, outside of the realm of Policy Gradients, one of our key contributions is better extending evolutionary-based methods (ex: Regularized Evolution) to scale better with continuous parameters, which we believe is novel.
> > > > > * Is it referring to only the standard class of Pointer Network methods (to allow defining a probability distribution over "typical" combinatorial spaces) from [(Le, 2017)](https://arxiv.org/abs/1611.01578)?
> > > > >   * In this case, we would like to point out the Pointer Network approach does not natively support continuous optimization (this can be seen in comments in the provided supplemental code, which is an official copy of the code used in [(Le, 2017)](https://arxiv.org/abs/1611.01578)). Hence we believe adding ES/CMA-ES variants are a sufficient contribution for extending [(Le, 2017)](https://arxiv.org/abs/1611.01578) to continuous spaces.
> > > > >   * Furthermore, Pointer Network-based approaches are not the clear winner; note that in Section 4.3.1 and 4.3.3, we do actually show that ES-ENAS (Evolutionary Controller) can sometimes outperform ES-ENAS (Pointer Network) on multiple occasions. This is corroborated by past works over pure combinatorial spaces $\mathcal{M}$ such as classic NAS, [Regularized Evolution has outperformed Pointer Network-based Policy Gradients, see Figures 3 and 4](https://arxiv.org/abs/1802.01548).

---

> > > > > > ### Comment · Reviewer_HAf2 · 2023-01-11
> > > > > > **Response**
> > > > > >
> > > > > > I used "Policy Gradient" as this term is used in your response: "... the notion of a J-function  and performs policy gradient updates." So I meant approaches like Akimoto et al. (2019). This is ready to use without tuning the choice of combinatorial optimizer. Moreover, as is mentioned before, this is as general as ES-ENAS in terms of applicable tasks. So I think Akimoto et al. (2019) is a baseline that the proposed approach should be compared to.
> > > > > >
> > > > > > "An analogous argument would be to say that well-known algorithms such as ARS/ES or even CMA-ES are not sufficient or novel contributions simply because they can be seen as "just Policy Gradients"" Yes, they are kinds of specific policy gradients. But I think that the usefulness of these specific types of policy gradients are supported by strong experimental results over baseline algorithms.

---

### Review · Reviewer_96o7 · 2022-12-03

**Summary Of Contributions:**

This paper develops an optimization method for hybrid search spaces, which combines the advantages of combinatorial optimization methods with evolutionary strategies (ES). The approach is an elegant extension of recent combinatorial and ES methods. The motivation is to apply the method in scenarios where the continuous-space dimensionality is large. The paper validates the approach in a synthetic hybrid benchmark, and then applies it to the problem of searching directly for NN weights for RL (i.e., w/o gradient descent), while simultaneously quantizing or sparsifying the networks. Here, the approach is able to successfully optimize both objectives, performing well compared to baselines.

**Audience:**

Yes

**Claims And Evidence:**

No

**Requested Changes:**

Changes critical to securing recommendation for acceptance:
- Clarify theory. When exactly does Theorem 1 imply a convergence speedup of ES over Mut, and how large is this speedup? The surrounding text suggests an exponential speedup, but this is not evident in experiments or from the Theory itself. This clarification will include fixing the misuse of \leq and \geq with big-Oh notation. Are there cases where the theory shows Mut is better than ES, e.g., when there is low smoothness and high concavity?
- Describe results and implications in the main text, i.e., not just in Figure captions and figures themselves.
- Address literature gaps.
- Present complete experiment results as outlined above, or describe why no such results are needed to support the main claims.

Other notes (not critical to securing recommendation for acceptance):
- The paper is surprised to find that RE stalls out, while PG keeps improving when using ES-ENAS. My guess is this is due to the fact that the power of RE comes from having a diverse population, but when combined with ES this diversity is severely limited, since ES is essentially local search. On the other hand, PG itself is local search, so it is not affected negatively in this way.
- First few paragraphs of Section 4.1 should be moved to previous section describing setup.
- "Thus an input (m, \theta) is evaluated" -> "An input (m, \theta) is then evaluated"
- The first sentence of Section 3.1 should mention that the problems are "hybrid variants" of existing benchmarks

**Strengths And Weaknesses:**

Strengths:

- The problem definition is well-motivated.
- The solution is elegant and theoretically-grounded.
- The result is a practical, modular, and relatively-simple answer to the open question of how best to combine discrete and continuous evolutionary optimization.
- The synthetic experiment provides a clearly-designed benchmark for investigating the validity of the method.
- The paper considers several important baselines and underlying methods, e.g., CMA-ES, PPO, Regularized Evolution (RE), NEAT, and Random Search.
- Intriguing future applications are highlighted such as circuit design and compiler optimization.
- The theory attempts to produce a fundamental result separating ES from Gaussian mutation, which would be of broad interest to the ML and optimization communities.

Weaknesses:
- The implications of the theory (Theorem 1) are not clear. It is not clear why it implies that mut needs exp(d) evaluations to take productive steps. It is not clear that quantity on the right-hand-side will always be greater than 1. The usage of \geq together with big-Oh is confusing and contradictory. Is it meant to be \omega? Use of \leq and \geq together with O(*) in the Theorem and Appendix are similarly inappropriate. It appears based on the theory that mutation can be more efficient than ES, depending on the smoothness or concavity of the problem. There is no analysis of whether the problems they consider have the correct smoothness and concavity properties.
- Descriptions of results and conclusions drawn from them are missing from the main text. Figure captions run over multiple pages and also do not fully describe results. This leaves the reader to interpret the results on their own, which is difficult, since all the experimental benchmarks and setups are novel.
- The method is framed in the context of literature and previous methods applied to searching for deep architectures or optimizing their weights. However, the experiments are only in a synthetic benchmark and small NNs for RL. Non-deep-learning methods may be better in such domains, e.g., the original NEAT and related work that combines NEAT with ES on small networks (e.g., CMA-TWEANN, Moriguchi and Horiden 2012).
- There are clear gaps in references to the literature: (1) "...including the well-known NEAT (Stanley and
Miikkulainen, 2002). Coincidentally, the neural architecture search (NAS) community has also
adopted a multitude of blackbox optimization methods..." This is not a coincidence, as some of the first NAS methods were based directly on NEAT and developed by the NEAT community, e.g., (Evolving Deep Neural Networks, Miikkulainen et al. 2017); (2) "...Tan introduced constrained optimization..." It may be that they introduced a specific type of constrained optimization for NAS, but did not introduce constrained optimization in general; (3) "...actively explored in the NAS literature..." What literature specifically?; (4) "...simultaneous gradients..." Mentioning other methods that successfully use simultaneous gradients could help increase reader confidence (e.g., GANs).
- The experimental results are incomplete: (1) The results in the synthetic benchmark are aggregated over all 19 problems, but the reader needs to see per-problem results (can be in Appendix) to know that the new method is not simply dominating on one or two of the problems (or at least give a metric for how many of the problems the new method is best on); (2) The RL results are only with one trial, and the domains have high variance, so it is hard for the reader to draw reliable conclusions; (3) Some baseline comparisons are missing, e.g., RE and PPO alone in the autorl domain (Is RE w/ d_cat = 0 equivalent to running deep-GA from Such et al 2017?).

---

> ### Author Response · Authors · 2022-12-05
> **Response to Reviewer 96o7**
>
> We thank the Reviewer for their careful review and multiple suggestions. We have incorporated them to improve the paper. Below are our direct answers to the Reviewer’s questions and requests.
>
> **Clarifying Theory:** We have edited the main section to be more clear about the theory. To answer directly, we summarize our proof steps:
> 1. Establishing standard assumptions of concavity on a blackbox objective $f$, and analyzing the expected 1-step improvements by ES and MUT algorithms given the same budget $B$.
> 2. Showing that an expected ES improvement $\Delta_{ES}$ can be lower bounded.
> 3. Showing that an expected MUT improvement $\Delta_{MUT}$ can be upper bounded.
>     * If $B$ is reasonably small (subexponential in $d$), then this upper bound on $\Delta_{MUT}$ turns out to be nontrivial.
>     * If $B$ is exponential in $d$, the upper bound becomes vacuous (because MUT will now do a brute force random search over the entire space), which is the only case in which MUT outperforms ES.
> 4. Using the bounds above to establish that ES is effectively improving faster than mutation when using reasonably sized compute (subexponential $B$).
>
> **Describe results and implications in main text:** Done, thanks for the suggestion.
>
> **Literature Gaps:**  We have modified much of our wording around those areas to be more careful around previous literature, and have added relevant citations, thank you for the pointers!
>
> **Experimental Results:** We have modified our paper on point (2) and explain why (1) and (3) are not needed:
> 1. Aggregation of Synthetic Results: The average of normalized optimality gaps ($\frac{f^* - \widehat{f^*} }{f^*}$) was the best way to convey our message; since normalization is used, no single function result will dominate over others. In general, there is no standard way to convey “average performance” in blackbox optimization literature. Some well-known alternative metrics were unsuitable:
>     * [Performance Profiles and Average Rank](https://arxiv.org/abs/cs/0102001) generally do not convey the numerical magnitude of the gap (which is emphasized by our theory)
>     * We did indeed find the same magnitude of performance gaps across every single function as well, but believed showing a plot for all 19 functions would be excessive.
> 2. As mentioned in Sec. 4.2, originally our RL results (e.g. in Table 1) used the 3-seeded standard, but for conciseness we omitted standard deviation counts (moved to Appendix C). We have put those back in (with Toeplitz/Circulant baselines copied from relevant papers).
> 3. Baselines: For Regularized Evolution (RE), we believe that Section 3 has already demonstrated enough that generally for functions involving high (>1000) continuous parameter count, RE’s sample complexity can be too high, even though (Such et al. 2017) have found similar methods suitable for specific domains. Section 4 is primarily an example of a downstream application, and hence we used more domain-specific baselines.
>     * PPO is also an unsuitable baseline as it requires a differentiable pipeline, while our work allows non-differentiable objectives.
>
> **Other Notes:** Incorporated suggestions, thanks!

---

> > ### Comment · Reviewer_96o7 · 2022-12-07
> > **Clarifying theory**
> >
> > Thanks for the updates and clarifications. I have a few lingering comments on the theory:
> >
> > 1. Is there an assumption that the condition number is independent of d? I.e., if it is Omega(d), then Theorem 1 gives no improvement. If this is a standard assumption, it should be made explicit in the paper with a reference.
> > 2. Assuming the condition number is constant, Theorem 1 gives at most an order d improvement, i.e., when B = O(1). The improvement decreases to O(1) as B increases to Omega(exp(d)). Although not technically incorrect, the text following the theorem is still misleading on this point: From the theorem, mut is at worst a linearly worse optimizer than ES, not exponentially worse. It may even be possible to characterize some of the empirical improvements of ES-ENAS in the BBOB problems as approaching linear with increasing d.

---

> > > ### Author Response · Authors · 2022-12-07
> > > **Response to Clarifying Theory**
> > >
> > > Great questions! Responses below:
> > >
> > > 1. When writing formal proofs in [convex optimization](https://web.stanford.edu/~boyd/cvxbook/bv_cvxbook.pdf), the condition number is treated as a constant independent of other parameters such as dimension $d$. We have referenced the Convex Optimization book in the updated draft.
> > >     * For certain classes of functions, it may be the case that the condition number would scale monotonically with $d$, but this is not guaranteed or even common. For instance, the sphere function's condition number is even completely independent of $d$, with similar properties for more general quadratic objectives.
> > >
> > > 2. We agree with the reviewer and made our wording more precise, but would also like to say that this also leads to downstream effects on convergence. Informally, it can be shown that if the expected improvement of a method at $x$ is $\Delta = r || \nabla f(x) ||^{2} $ where is $r < 1$ is some constant, then running such a method $k$ times will lead to $f(x_{k}) \approx f(x^*) k r $, which will lead to needing $k=O(1/r)$ iterations to reach optimality at $f(x^*)$.
> > >     * As you mentioned, for the MUT operation, when using subexponential $B$, we will have this ratio $r_{MUT}$ upper bounded by an order of $d$, and hence the optimization process will take $d$ times longer than ES. We believe that for high dimensions (e.g. $d=1000$), this can be a serious issue however, demonstrated in Figure 2 (note the log scale on the y-axis).

---

> > > > ### Comment · Reviewer_96o7 · 2022-12-08
> > > > **Followup on theory**
> > > >
> > > > 1. Ah, ok, thanks for the pointer. I see now that the paper notes that the result "does not cover the case for non-convex objectives", which is motivation for running on BBOB, which includes highly non-convex functions. The BBOB paper (Hansen, et al '09) says the testbed functions can have condition number 10^6, which is higher than the dimensionality d considered in experiments in the current paper. In this case, Theorem 1 gives no advantage for ES, but suggests an advantage for mut could be possible. This is one reason why separately reporting results for the well-conditioned and ill-conditioned BBOB problems could give more insight into when ES-ENAS should be preferred. Just from the theory, we may think the positive results of ES-ENAS on BBOB are actually surprising.
> > > >
> > > > 2. Yep, I agree, this up-to-order d speedup can have huge practical significance for large d. Since the section is called "Curse of Continuous Dimensionality" and highlights d vs exp(d) differences between ES and mut, the reader may still conclude that ES can make convergence exponentially more efficient. I think it would be clearer to start the section by making the point: the section shows that ES can have up to an order d speed-up over the other methods, which can have huge practical implications when d is large.

---

> > > > > ### Author Response · Authors · 2022-12-10
> > > > > **Thanks for the suggestions!**
> > > > >
> > > > > 1. Thank you for the suggestion on separating the BBOB results over different condition number classes - we will definitely incorporate these in the final version of the paper. In general, we found the convergence curves to be very similar across all BBOB functions (even for non-convex objectives with high condition numbers), which suggests that there are also additional benefits to using ES even in non-convex settings, outside of convergence rates.
> > > > >
> > > > > 2. This is a great suggestion. We have incorporated the suggestion of an order $d$ speedup into Section 3 in the updated draft.

---

> > > > > > ### Comment · Reviewer_96o7 · 2022-12-16
> > > > > > **Remaining confusion in theory**
> > > > > >
> > > > > > Thanks for the discussion, and updates so far, including some updates surrounding the $O(\ldots)$ notation.
> > > > > >
> > > > > > There are still some places where its use is confusing, and looking more closely at these places, they appear to not support the conclusion in Theorem 1:
> > > > > >
> > > > > > 1. $\Delta_{ES}(\theta) \geq O(\ldots)$ means that $\Delta_{ES}(\theta)$ is at least as big as a function that can be arbitrarily small.
> > > > > >
> > > > > > 2. $\Delta_{MUT}(\theta) \leq \frac{\ldots}{O(\ldots)}$ means $\Delta_{MUT}(\theta)$ is no bigger than something that can be arbitrarily large.
> > > > > >
> > > > > > Together these cannot lead to a meaningful bound. It's possible that these $O$'s can be made to be $\Theta$'s by improving the bounds used for the proofs in the Appendix. Is this improvement possible? Or have I somehow misinterpreted the use of $\geq$ and $\leq$ with $O(\ldots)$?

---

> > > > > > > ### Author Response · Authors · 2022-12-17
> > > > > > > **Response**
> > > > > > >
> > > > > > > Thanks for the input; We've edited Theorem 1 and the Appendix to follow the more strict [notation from Big-O](https://en.wikipedia.org/wiki/Big_O_notation) (to avoid using $\ge$ or $\le$ with big-O's) and have updated the draft.

---

> > > > > > > > ### Comment · Reviewer_96o7 · 2022-12-19
> > > > > > > > **Followup**
> > > > > > > >
> > > > > > > > Thanks for the updates. It looks like at least Eq 11 in the Appendix still requires a similar change.

---

> > > > > > > > > ### Author Response · Authors · 2022-12-19
> > > > > > > > > **Response**
> > > > > > > > >
> > > > > > > > > Thanks for the careful reading - we've edited to use $\Theta$ in Eq (11) and checked that the earlier text citing (Nesterov and Spokoinoy, 2017) use $O(\cdot)$ correctly to denote upper bounding.

---

### Author Response · Authors · 2022-12-05
**General Comment to All Reviewers**

We thank all the reviewers for their time in writing their feedback! We have edited our paper (changes in **red**).

In general, we have modified our claims to be more precise, and to avoid any potential overstatements following TMLR guidelines. We agree that there is a vast literature on blackbox optimization, NAS, evolutionary algorithms, and AutoML, and we definitely do not intend to compete against all such methods, but rather claim ES-ENAS’s merits (namely generality, simplicity, and efficiency).

Please let us know if anything else is needed, and we would be happy to discuss further.

Thank you!

---

### Decision · Action_Editors · 2023-01-09

**Recommendation:** Reject

**Comment:**

As indicated above, the main concern is with the level of support for the claims made.  Despite the decision, this work may be publishable with suitable non-minor editing. Either (i) the main claims of the paper could be made significantly more modest while ensuring that they are always adequately supported, or (ii) further supporting theory or experiments to ensure that all claims made are clear and convincing.

(Note: I acknowledge that a recent request for clarification to a reviewer was not answered, but since the decision was unanimous, I decided to proceed as is.)

**Audience:**

This work should be of interest to some researchers working on Neural Architecture Search.  The level of interest to those working on general black-box optimization is currently less clear.

**Claims And Evidence:**

The reviewers generally agree that the problem is well-motivated and of interest.  However, following the author response and discussion, a consensus was reached that the theory and experiments are not sufficient to support the main claims made.

For example, the reviewers mentioned the following:
-  The abstract suggests solving a key open problem, but after reading the paper the reviewer remains unclear on when the method is suitable and should be used.
- The paper is introduced as tackling general-purpose black-box optimization, but the paper is then much more specific to NAS.
- While the algorithm does seem to be applicable in more general scenarios, its usefulness in such scenarios is not sufficiently supported
- The theory is appreciated but is restrictive due the convexity assumptions, so does not eliminate the reviewer concerns